# Hydropower capacity factors trending down in the United States

Sean W. D. Turner [1] ✉, Ganesh R. Ghimire [1], Carly Hansen[1], Debjani Singh[1] & Shih-Chieh Kao [1]

The United States hydropower fleet has faced increasing environmental and regulatory pressures over the last half century, potentially constraining total generation. Here we show that annual capacity factor has declined at four fifths of United States hydropower plants since 1980, with two thirds of decreasing trends significant at $p < 0.05$. Results are based on an analysis of annual energy generation totals and nameplate capacities for 610 plants (>5 megawatt), representing 87% of total conventional hydropower capacity in the United States. On aggregate, changes in capacity factor imply a fleetwide, cumulative generation decrease of 23% since 1980 before factoring in capacity upgrades—akin to retiring a Hoover Dam once every two to three years. Changes in water availability explain energy decline in only 21% of plants, highlighting the importance of non-climatic drivers of generation, including deterioration of plant equipment as well as changes to dam operations in support of nonpower objectives.

Conventional hydropower plants in the United States provide clean, renewable electricity with negligible marginal production costs and a host of flexible generation and ancillary services to the grid[1–3]. Achieving a decarbonized electricity supply will depend on the ability of the hydropower fleet to uphold historical levels of energy generation while facilitating increased penetration of variable renewable energy (VRE) technologies through its unique load balancing capabilities[4,5]. Yet this may prove challenging for a variety of reasons. Hydropower plants depend on river flows and must operate within the confines of site-specific rules and regulations that govern the storage of water behind the dam and the pattern of water release through penstocks and spill gates[6–8]. Such constraints can be multifaceted, ranging from seasonal guide curves designed to balance flood buffer and water conservation in the reservoir, to sub-daily requirements on the spill and turbined release to promote fish passage, downstream amenity, safety, and water quality[9]. Meanwhile, new market mechanisms are rewarding hydropower producers to operate below capacity to allow ramp-up when needed. Incentives for flexible operations may enhance hydropower profitability and facilitate the integration of VRE technologies, but tend to reduce total annual electricity generated from participating plants[10]. Thus, geophysical change (climate or hydrological change), sociopolitical trends (e.g., increasing emphasis on flood control, or environmental preservation), and electricity market developments all have the potential to reduce total generation from hydropower over the long term. The relative importance of these disparate phenomena and their long-term impacts on annual generation across the hydropower fleet remain unexplored in a US-scale study.

The prospect of a long-term decline in annual hydropower generation matters. Today in the United States, hydropower contributes approximately 6% of total electricity generation and 32% of utility-scale renewable generation (based on 2019–2022 data[11]). Although this share is declining as wind and solar technologies continue to expand[12], hydropower remains an essential contributor of generation in the Pacific Northwest (63% of average annual electricity generation in the Washington-Oregon-Idaho tri-state area), California (13%), Southeastern United States (9% in Tennessee-Alabama), and the Northeastern United States (21% in New York–Vermont–Maine–New Hampshire)[13]. Such is the importance of hydropower to these regions that drought-caused losses in hydropower generation can compromise the reliability of electricity supply[14], leading to replacement generation from fossil-fuel plants. The latter results in higher grid costs[15] and increased greenhouse gas emissions[16]—even causing a detectable bump in air pollution downwind of ramped-up coal plants[17].

[1]Environmental Sciences Division, Oak Ridge National Laboratory, Oak Ridge, TN 37831, USA. ✉e-mail: turnersw@ornl.gov

Yet despite the recognized importance of these short-lived effects, the prospect of more enduring losses in hydropower generation has been studied primarily in forward-looking climate impact assessments (e.g., refs. [18,19]) rather than retrospectively. A multidecadal, plant-level, retrospective analysis of long-term trends in generation can help explain how hydropower generation has been affected by both climatic and non-climatic drivers, including changes to catchment hydrology, reservoir operations, environmental regulations, and power grid needs. Understanding how these diverse drivers affect annual hydropower generation will be important for estimating the fleet's future capabilities and informing capacity expansion and grid reliability models used to study the performance of current and proposed power grid configurations in the United States[20–22].

To explore the evolution of annual generation from hydropower in the US over the last four decades, we create and analyze time series of generation (MWh), installed nameplate capacity (MW), and annualized capacity factors (CF) for 610 out of 808 US hydropower plants with a nameplate capacity of 5 MW or greater (covering 90% of installed capacity in the >5 MW category and 87% of overall US conventional hydropower capacity) over the period 1980–2022. Annual CF is defined as total generation divided by the maximum possible generation implied by the nameplate capacity of the plant within the calendar year, consistent with most formal definitions. Use of annual CF to measure trend (in addition to generation directly) removes the effects of any capacity upgrades or unit retirements over the last four decades. Importantly, a decreasing trend in annual CF may not necessarily mean that a hydropower plant is running less often. It may also suggest lower conversion from flow to energy (due to impaired generator efficiency, lower average head levels, etc.) or reduced flow through turbines and thus lower output while running. Using a robust statistical model that simulates annual CF from historical reservoir inflow at each plant, we determine the extent to which observed trends in hydropower CF are a function of changing water availability. The use of both natural and gauge-assimilated historical streamflow to drive this model allows us to separate the influence of climate from other drivers of hydrological change.

We report a pattern of widespread decreasing CF, with four-fifths of plants showing a decreasing trend and a resulting cumulative net loss of 13% generation fleetwide since 1980–despite capacity upgrades at more than half of analyzed plants. After accounting for and removing the effects of capacity upgrades, the cumulative loss in generation since 1980 is 23%. Water availability seems to be the predominant driver of CF trend for only 21% of plants that show long-term CF decline; one-third of plants show long-term CF decline despite increasing water availability. To account for the majority of cases where water availability is not a prominent driver of CF trend, we suggest a range of possible drivers, separating these into categories of external environmental phenomena, infrastructural change, and operational change. We speculate that declining hydropower CF is characteristic of an industry that has traded a modest portion of generation for an improved ability to meet evolving water and power grid needs. Whether hydropower CF will continue to decline into the middle of the 21st century remains an open question.

## Results

### Widespread decreasing trend in hydropower capacity factors

Approximately four-fifths of US hydropower plants with nameplate capacity >5 MW show a long-term decline in annual CF over the last four decades (i.e., 482 out of 610 plants studied show a negative trend in CF for the period 1980–2022) (Fig. 1a). Plants with declining CF trend account for 85% of installed capacity and 84% of average annual generation across the sample. About two-thirds of plants showing CF decline (318 out of 482) are associated with a trend that is statistically significant at $p < 0.05$ (see "Methods" section for the approach to establishing statistical significance). Fewer than one-fifth of the 128

plants with increasing CF show trends that are statistically significant at $p < 0.05$. There are therefore about thirteen times as many plants that experienced a statistically significant decrease as those that experienced a statistically significant increase in CF over the period 1980–2022. Across all plants studied, the median trend in capacity factor is -2.6 percentage points per decade (PPPD), or −0.26 percentage points per year. Trends range from −11.3 PPPD at the 5th percentile to +1.9 PPPD at the 95th percentile of plants studied.

Declining capacity factor trends are particularly common at the largest plants by nameplate capacity. To illustrate, 84% of plants with nameplate of 100 MW or more show a negative trend, compared to 78% of plants with nameplate less than 100 MW. All ten plants with nameplates>1000 MW show a negative trend in capacity factor (seven of these are statistically significant at $p < 0.05$). These generation losses at the largest plants contribute a significant net effect: the overall change in capacity factors, across all plants studied, implies a net rate of decline of 14.5 TWh generation per decade (before accounting for effects of capacity upgrades). These losses translate into a fleetwide, cumulative generation decrease of 23% since 1980 (assuming static capacity). This net long-term historical trend in hydropower utilization between 1980 and 2022 is equivalent to the retirement of a Hoover Dam once every two to three years (Hoover Dam produced an average of 3.6 TWh energy per year over the last decade).

Declining hydropower CF is observed throughout the nation but is most prevalent in the West. In the three major western hydrologic regions—Pacific Northwest (PNW, USGS Hydrologic Unit Code, or HUC, 17; Fig. 1b), California (HUC 18; Fig. 1c), and Colorado River Basin (CRB, HUCs 14 and 15; Fig. 1d)—most plants exhibit significant long-term declines in CF, along with notable negative trends in regional CF (i.e., total annual hydropower generation in the region divided by the total nameplate capacity across all plants in the region). California is associated with large interannual variability in water conditions (and thus annual CF), yet also exhibits a clear pattern of long-term CF decline. Although large declines in annual hydropower CF in California are driven in part by severe multi-year droughts in years 2013–2015 and 2020–2021, the shorter, 30-year period of 1980–2009 is also associated with similar prevalence and magnitude of decreasing trends (see Supplementary Information, Fig. S1). For the PNW, where hydropower accounts for close to two-thirds of total electricity generation (and about one third of total hydropower generation in the U.S.), the overall implied net loss in hydropower generation is 6.2 TWh per decade, equal to just over 5% of the region's annual hydropower output and 4% of total annual regional output across all generating technologies. Given the large contribution of PNW hydropower to overall U.S. hydropower generation, the CF trend in this region has a significant bearing on the national CF trend.

In contrast to western regions, the southeast US shows no obvious regional CF trend (Fig. 1e). Relatively few plants in this region have experienced significant long-term trends in either direction. Exceptions include a cluster of plants in southern Alabama and a string of plants on or near the Ohio River at the northern edge of the region (all with negative trend significant at $p < 0.05$). In the northeastern US, trends vary significantly across plants, although the dominance of Robert Moses Niagara results in an overall decreasing pattern of CF here (Fig. 1f). The outsized effect of this hydropower plant, which generates with relatively high CF compared to other plants in the region, is also evident in high regional CF.

Capacity upgrades resulting from additional or replacement generating units at a plant mean that declines in CF shown here do not necessarily translate to declines in actual generation. For instance, declining annual CF with increasing nameplate capacity can accompany annual generating output with no long-term trend. To analyze the effect of capacity upgrades on the results presented here, we perform a complementary analysis on generation directly (see Supplementary Information, Fig. S2). We find that despite some 342 plants out of the

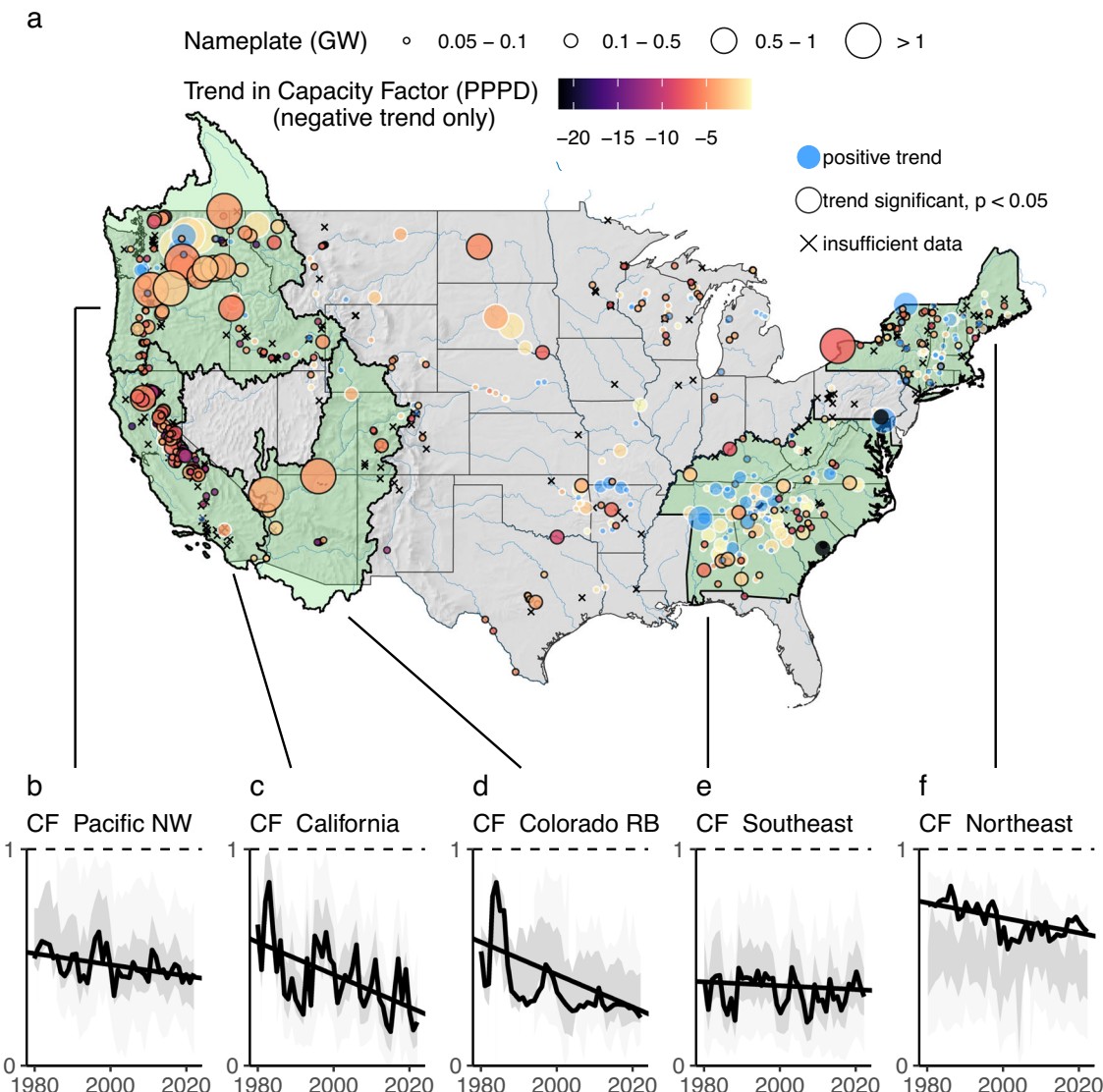

**Fig. 1 | Annual capacity factors trending down for most plants and regions.** **a** shows 610 analyzed CONUS plants with > 5 MW nameplate in 2021 (~90% of US hydro capacity) (198 plants with insufficient data are marked with an x). Point fill gives a 43-year (1980–2022) linear capacity factor trend in units of percentage points per decade (PPPD). **b**–**f** give regional capacity factors for the Pacific Northwest (HUC 17), California (HUC 18), Colorado River Basin (HUCs 14 and 15), Southeast (TN, AL, GA, SC, NC, KY, WV, MD, VA), and Northeast (NY, VT, ME, NH, CT, MA, RI). Gray-shaded ribbons give an interquartile range (dark gray) and the 5th to 95th percentile range (lighter gray) of annual capacity factors across all plants within each region. In (**f**), high CF at the region's largest plant causes regional CF to lie outside the interquartile range (regional annual CF is total regional generation over total regional capacity–not an average CF across plants).

610 plants analyzed having had capacity added at some point in the last 43 years, generation has declined at approximately 70% of plants overall. In other words, while almost 60% of plants have added some capacity during the last 43 years, only 30% of plants have increased generation over this time period. Of the 298 plants with declining CF and which have had capacity added at some point over the last 43 years, 224 plants (75%) experienced a decline in generation despite having upgraded their capacity. Nonetheless, capacity upgrades significantly temper the aggregate effect of losses in CF when analyzed at CONUS and regional scales (Table 1). At a fleetwide (CONUS) level, the actual rate of generation decline since 1980 is about 8.2 TWh per decade (cumulative loss of 13% generation since 1980), or half of the generation decline implied by CF change alone.

**Water availability cannot fully explain the observed CF decline**
While being an important driver of CF trend at many plants, water availability cannot fully explain the observed widespread long-term CF

decline (Fig. 2a). The following analysis is based on a sample of 362 plants (herein "modeled plants") for which gauge-assimilated reanalysis streamflow data are accurate enough to simulate annual CF (see "Methods" section). Plants in this sample account for 70% of the total US-installed hydropower capacity. Results from the CF trend analysis reported above show a negative CF trend for 288 of these 362 modeled plants (i.e., roughly four-fifths of modeled plants with the negative trend–in tune with the larger sample of 610 plants). For each modeled plant, the proportion of the trend explained by water availability is computed as the plant's CF trend modeled with historical streamflow divided by the CF trend observed. For example, a plant with a negative CF trend of 2.0 PPPD and with a modeled negative CF trend of 1.5 PPPD is computed as having 75% of its trend explained by the change in water availability.

Across the 288 modeled plants with negative observed CF trend, removal of the portion of trend explained by water availability results in only a modest adjustment of the median CF trend from −2.9 PPPD

**Table 1 | Wide regional variation in overall CF and generation decline**

| Region | Trend in annual CF, percentage points per decade (PPPD) | Implied rate of change in annual generation (assuming no capacity upgrade/retirement), TWh per decade [cumulative % change since 1980] | Actual rate of change in annual generation (accounting for capacity upgrade/retirement), TWh per decade [cumulative % change since 1980] |
|---|---|---|---|
| CONUS | −2.9 PPPD | −14.5 [−22.8%] | −8.2 [−12.9%] |
| Pacific NW | −2.6 PPPD | −5.3 [−18.5%] | −2.7 [−9.6%] |
| California | −7.2 PPPD | −4.0 [−46.3%] | −3.4 [39.4%] |
| Colorado Basin | −3.8 PPPD | −1.1 [−35.1%] | −0.9 [−28.8%] |
| Northeast | −4.5 PPPD | −1.3 [−18.8%] | −0.2 [−3.0%] |
| Southeast | −1.0 PPPD | −1.0 [−13.0%] | −0.14 [−1.9%] |

Regional CF is computed by first totaling annual generation across all plants, then dividing by total nameplate across plants (i.e., regional CF is not an average of CF trends across each region). The implied generation trend is computed assuming the 1980 nameplate through the entire period 1980–2022 (implied generation = CF multiplied by nameplate generation). Actual generation is taken directly from the observed annual reported generation. The actual generation change illustrates the impact of capacity upgrades on rates of generation decline.

(−9.6 and −0.2 at the 5th and 95th percentiles, respectively) to −2.5 PPPD (−7.3, +0.5). Only 38 of 288 plants showing negative CF trend are found to be strongly driven by water (i.e., water availability explaining more than 80% of observed CF trend), and only one-fifth of negative trends (62 out of 288 plants) are predominantly driven by water (i.e., water availability explaining 60% of trend or more). There are 156 plants for which water availability explains less than 20% of the observed CF trend. More strikingly, there are 117 plants (one-third of the 362 analyzed plants) that show a negative CF trend despite increasing water availability (thus 0% of the trend is explained by water availability). A tenth of analyzed plants show both significant negative CF trend ($p < 0.05$) and increasing water availability. Interestingly, 54 out of 74 plants with positive CF trends are predominantly driven by water. In other words, most plants with long-term positive CF trends benefit from increased water availability, whereas plants experiencing long-term negative CF trends require explanations beyond water availability to account for the data.

Although the influence of water availability on CF trend varies throughout CONUS, some regional patterns are evident. Water availability is a strong driver of trend in the southwest and other parts of the west, including California. Examples include Shasta in northern California (Fig. 2b), and Hoover Dam (Fig. 2c) on the Colorado River, where prolonged, multi-year drought has impaired hydropower generation in recent years[23]. Other regions where water availability explains 40% or more of CF trend are distributed throughout CONUS, and include the northeast (e.g., Robert Moses Niagara, Fig. 2d), various parts of the Columbia River Basin (e.g., Bliss on the Upper Snake, Fig. 2e, and Wanapum in the Middle Columbia, Fig. 2f), the Upper Missouri (e.g., Garrison, Fig. 2g), Texas, and the Southeast. Isolated examples where water availability explains 20–40% of CF trend include Millers Ferry on the Alabama River (Fig. 2h) and Dardanelle on the Arkansas River (Fig. 2i). Plants with a statistically significant negative trend in CF that cannot be attributed to water availability are clustered along the lower Snake (e.g., Lower Granite, Fig. 2j), the Cascades in Oregon (e.g., Foster, Fig. 2k), throughout the Southeast, and in a selection of other basins. Fort Peck in the Upper Missouri (Fig. 2l) and Kentucky Dam (Fig. 2m) are examples of plants with negative CF trends despite increasing trends in water availability.

Most CF trends driven by a change in water availability can be attributed to a change in climate rather than other influences on water availability (e.g., land use change, human water consumption) (Fig. 3a, b). This finding is based on a comparison of modeled CF using gauge-assimilated streamflow with modeled CF using natural streamflow unaffected by other drivers of hydrology. We find that excluding non-climatic drivers with the natural flow setting does not markedly change overall results on the influence of water availability on CF trends. While actual CF trends are negative for 288 of 362 modeled plants, the CF trend simulated using natural water availability is negative for only 134 of 362 modeled plants (Fig. 3a). In other words, if climate change was the only driver of annual hydropower generation in the United States, one would expect to see only 37% of plants with a decreasing CF trend over the last forty years, rather than the observed 80% of plants with a decreasing CF trend. Including non-climatic drivers of water availability through streamflow gauge assimilation improves the capture of observed CF trend only marginally relative to the natural flow model, leading to 181 plants (50% of modeled plants) with a negative modeled CF trend. Filtering for the 152 plants with a negative trend in CF that is statistically significant $p < 0.05$ gives a similar outcome (Fig. 3b).

The minor overall discrepancy between CF trends modeled with natural versus gauge-assimilated streamflow is the result of large discrepancies between these two settings at a small number of plants. For 78% of modeled plants, the CF trend modeled with natural streamflow is of similar magnitude to the CF trend modeled with gauge-assimilated streamflow (within 10%). The exceptions, which indicate an important influence of other hydrological drivers in a handful of catchments in CONUS, are highlighted in Fig. 4. Possible drivers of decreasing water availability in these basins include land use and land cover change (affecting rates of runoff generation and evaporation) and changing patterns of human water consumption, including surface water and groundwater extraction for irrigation, which is a major water use category in some of the affected basins (e.g., Upper Klamath, Mid-to-Upper Snake, Upper Red River, Neosho). Changing patterns in the transfer of water into or out of these basins may also be responsible for some changes observed. Industrial, municipal, and electricity sector water use accounts for more significant portions of overall water consumption in eastern basins; changes in these sectors may partly explain changing water availability absent from natural streamflow simulations. In addition, uncertainty in hydrologic modeling and streamflow observations that form the assimilated streamflow may also contribute to the differences. Although interesting, these cases are the exception; in general, non-climatic influence on water availability upstream of hydropower dams is not a key factor in widespread, national-scale CF decline.

## Discussion
### Possible causes of long-term CF decline
CF trends that cannot be explained by changes in annual water availability could be caused by a myriad of other drivers. To organize this discussion, we identify four possible categories of drivers for long-term CF change: (1) external environmental phenomena (which include annual water availability as analyzed above), (2) infrastructural change, (3) dam operational change relating to evolving river and reservoir needs, and (4) dam operational change relating to evolving power grid needs (Table 2).

External, long-term environmental phenomena that are beyond the control of dam operators are a primary driver of change in

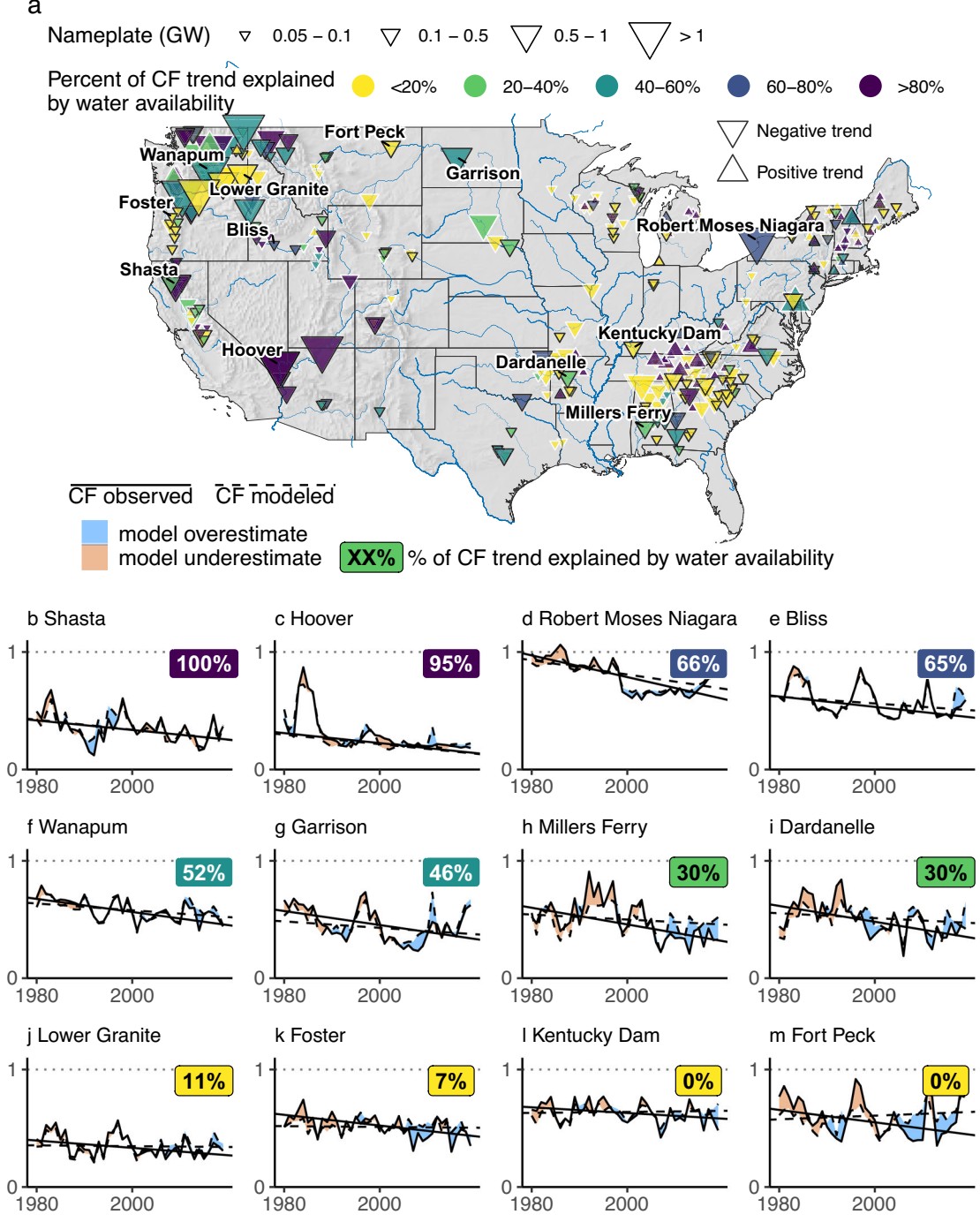

**Fig. 2 | Change in water availability cannot explain all trends in the annual capacity factor. a** gives the percentage of trend explained by water for each plant; plants with annual CF trends that are statistically significant at $p < 0.05$ are highlighted with a solid black outline. Each of **b–m** gives an example of actual and modeled generation at a plant. Together, these examples show the widely varying influence of water availability on the CF trend, from very strong (explaining all CF trends) to very weak (explaining no CF trends). In each panel, the time series and trends represented with broken lines are those determined with the CF modeled directly from water availability.

hydropower generation, as we have seen. Annual water availability for hydropower generation has changed significantly in many U.S. regions, leading to a decline in CF in the southwest, California, and elsewhere. Changes in the seasonal distribution of water availability could also be important; anthropogenic climate change has been linked to intensification of streamflow seasonality in unregulated, snow-dominated watersheds of the United States, driven by changes to precipitation patterns and warmer winter temperatures (affecting snowpack)[24].

A stronger seasonal signal in affected watersheds means more water flowing into reservoirs in springtime, potentially leading to more spilled energy, particularly at run-of-river facilities where high flows that exceed turbine capacity cannot be stored for energy generation later. This may explain some of the declining CF trends at large run-of-river dams in the Columbia River Basin, although here (and elsewhere) any intensification of the natural flow regime is dampened by large dams (powered and non-powered) further upstream in the basin[25].

a

All modeled plants (n = 362)

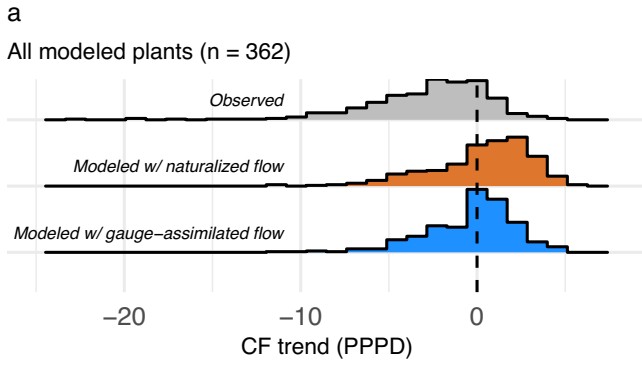

b

Modeled plants with negative trend* in observed CF (n = 152)

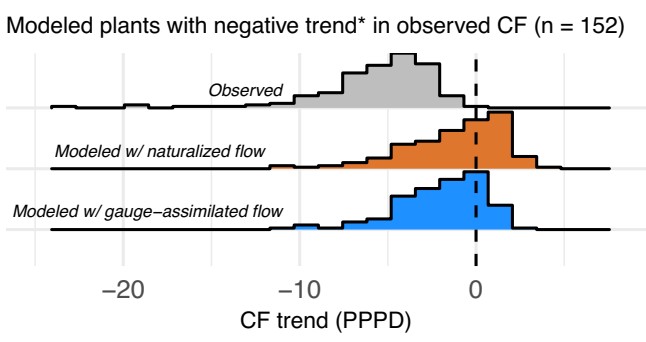

**Fig. 3 | With no other factors in play, a change in water availability would result in increased CF for most hydropower plants studied. a** gives distributions across all 362 modeled plants, while **b** is filtered for plants with negative annual CF trend (*statistically significant at $p < 0.05$).

A stronger seasonal flow signal could also prompt operators of storage dams to lower flood control guide curves, leading to reduced hydraulic head levels and thus less generation per unit turbined flow. Increased frequency of controlled and uncontrolled spill is a further possible effect of changing hydrological extremes that could reduce a plant's annual generation.

Although we lack conclusive evidence on whether observed CF declines are driven in part by changing within-year hydrology and high-flow events, we can test for significant trends in these hydrological features. We thus perform an analysis of the trend in five metrics computed from daily streamflow across a sample of plants (see "Methods" section). The metrics are 90th percentile of daily inflow (computed annually), annual maxima of daily, 7-day (rolling), and 30-day (rolling) inflow, and proportion of annual inflow arriving on the wettest month. Across 107 plants analyzed (i.e., those showing significant CF decline not well explained by annual water availability trend), the evidence for significant change in the flow regimes is limited (see Supplementary Information, Fig. S3). About three-quarters of these plants show no significant increase in any of the flow metrics analyzed. No plants in the Northwest (WA, OR, ID) show a significant trend in these metrics. Plants in the state of New York are associated with an increasing trend in 90th percentile of daily inflow, though not in annual maxima of daily flow. Importantly, the presence of a trend in these metrics does not necessarily imply that the flow regime trend is the driver of CF decline. The general absence of significant trends in these metrics indicates that changing within-year hydrology is likely not a major driver of hydropower CF decline in the United States. Analysis of flow bypassing the turbines (spill) at individual sites would provide stronger evidence as to the importance of seasonal and peak flow changes at plants. At present, these spill data are unavailable for a large sample of US hydropower plants.

Sediment build-up is another external environmental driver that could affect annual hydropower generation if the associated loss of storage leads to more frequent spills. Attributing observed CF trends to this potential driver is challenging. Data on US reservoir sedimentation are piecemeal and storage loss estimates across a large sample of dams are unavailable[26]. A recent estimate suggested that sedimentation has caused overall U.S. reservoir storage to decline from 850 Gm3 in the 1980s to 810 Gm3 today[27], although rates of storage loss will vary dramatically across dams as a function of upstream catchment characteristics and change. Importantly, a significant loss of storage does not necessarily lead to reduced hydropower generation at the plant, since sediment build-up affects neither hydraulic head nor inflowing water directly. Nonetheless, further data collection, including a large sample of reservoirs bathymetry surveys taken over time (or remotely sensed equivalent), and detailed modeling would be required to rule out sedimentation as a primary driver of CF trend.

Infrastructural change is the second category of possible drivers of the CF trend identified here. This category includes both infrastructure deterioration and infrastructural improvements. An example of the former is wear-and-tear, which may lead to increased unit downtime for repair as well as lowered efficiency. An example of the latter is capacity uprating, which would increase average generation but may reduce capacity factor if water becomes constraining at higher ratings. Our data suggest that capacity additions have not been a significant driver of CF trends observed in this study. Nameplate capacities have remained constant or have marginally declined for about half of the studied plants; only 37% of plants showing significant CF decline are associated with a nameplate upgrade of more than 5% during the study period. Of those that have been upgraded and for which significant CF decline is observed, we identify just six plants with evidence for a historical capacity increase coinciding with or preceding a downshift in output relative to water available (see Supplementary Information, Fig. S5). We thus suspect that nameplate upgrade is a peripheral rather than a major driver of CF declines observed in our study.

There is some indirect evidence that infrastructure deterioration could have contributed to long-term CF decline. This comes in the form of mandatory outage reports provided by plant operators to the North American Electric Reliability Corporation (NERC) via its Generating Availability Data System (GADS), summarized by ref. 28. Their analysis finds that hydropower units are experiencing more downtime for both planned outages and forced outages. Planned outage refers to routine maintenance, which may be required at an increasing frequency for older generators. Reference 28 reports that the average planned outage has increased by 41% over the last decade for units >100 MW. For smaller units, planned outages have almost doubled over this period. The most common reasons reported for these outages relate to turbine and generator component issues. Failures in turbine and generator components also account for 69% of generation lost due to forced outages. Aging infrastructure was also identified by the U.S. Army Corps of Engineers (a major plant operator throughout the United States) as a primary reason for a decline in plant availability from an average of 95% in 1987 to 87% in 1995[29]. Importantly, hydropower plant downtime does not necessarily affect the annual capacity factor as planned outages can be scheduled for times when the turbine is not running. If water is held in storage during a planned or forced outage, then it can used to generate power later. A lack of plant identifiers in the outage data prevents us from analyzing the extent to which these factors have influenced the hydropower CF trends reported in our study. Generator age appears to be unrelated to CF decline (see Supplementary Information, Fig. S4), although our analysis on this factor is inconclusive since the reported age of the generator does not account for refurbishments since installation.

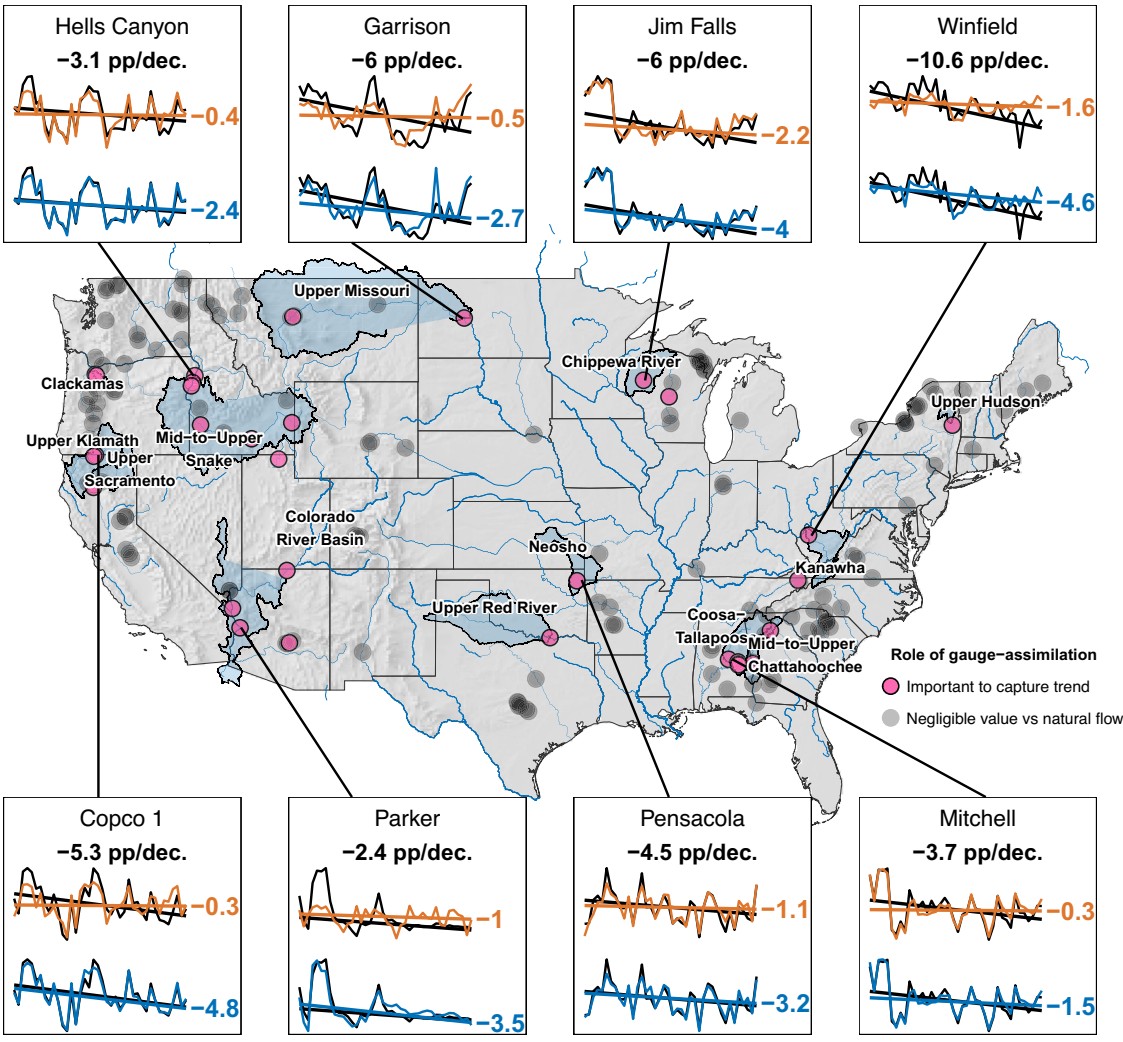

**Fig. 4 | Gauge-assimilated flows significantly improve the capture of trend relative to the natural flow in just a few plants.** Highlighted river basins (blue shading) contain one or more such plants (pink points). Spark lines for eight such cases show the observed trend in annual CF (black, bold type) and modeled trend using natural streamflow (orange) and gauge-assimilated (blue) streamflow (the slope of the trend line lies to the right of each sparkline, highlighting significant improvement in annual CF trend capture with the gauge-assimilated flow).

A possible major driver of long-term CF decline is a change in reservoir operations. We divide operational change into two subcategories: operational change to support river and reservoir needs (non-power operations), and operational change to support power grid needs. Non-power operations may include river ecology enhancement, navigation, recreation, water supply (including municipal, industrial, and irrigation water), and flood control objectives. Operational changes for these objectives can affect hydropower generation by increasing spilled energy, either directly (e.g., desired spill to promote fish passage) or indirectly (e.g., undesired spill resulting from changes to reservoir control curves). Although one can find anecdotal information on such changes for high-profile dam projects, a lack of consistent, quantified data on such operational changes prevents us from linking CF trends to operational change across a large sample of dams. Determining the possible timing of such changes in operations is also challenging since these may or may not occur as part of hydropower relicensing.

There may be clues in the generating behavior of hydropower plants that indicate possible operational change. One indication of operational change might be the presence of an abrupt shift in annual generation relative to available water. Such a shift would be unlikely to arise as a result of external environmental phenomena, like sediment build-up, or mechanical wear-and-tear; these drivers would cause a more gradual decline in capacity factors. Operational change, in contrast, could present as a downshift to a new and sustained pattern of energy generation, particularly if related to regulatory compliance such as a FERC relicensing agreement. The Electric Consumers Protection Act (ECPA) of 1986 increased the importance of environmental considerations in the hydropower licensing process for non-federal dams (which constitute most hydropower plants in the United States), with an expanded role for State and Federal fish and wildlife agencies. By the early 1990s, evidence was emerging that operational changes at dams relicensed post-1986 were more often detrimental to plant generation than not[30]. Given the large interannual variability of hydropower generation and the possibility that operational changes may have occurred at various points over the last four decades, clean separation of gradual trends from abrupt shifts in our CF time series is difficult. Nonetheless, a simple test for the presence of a single shift in the mean over the 40-year period (see "Methods" section) suggests that such shifts are commonplace, often presenting stronger statistical evidence than gradual trends in the data. In this study, there are 85 plants with a significant negative CF trend for which water availability explains 40% or less of that trend. About half of these plants exhibit a shift in the mean CF that presents with stronger statistical evidence

**Table 2 | Summary of potential drivers of long-term hydropower generation and capacity factor decline**

| Driver | Description | Mechanism for reducing annual CF |
|---|---|---|
| **External environmental phenomena** | | |
| Decline in annual water availability | Decline in inflow to reservoirs caused by climate change or upstream catchment process change | Reduced water available for release to penstock; reduced reservoir head levels |
| Shift in the seasonal flow signal | Change in the timing of water inflows to the reservoir, with a stronger flood season | Stronger flood leads to increased spills and thus a smaller proportion of water used for power generation |
| Enhanced reservoir surface evaporation | Warming temperatures promote an increased loss of water from storage | Reduced water availability for generation |
| Sedimentation | Accumulation of sediments in hydropower reservoirs | Loss of water storage capacity, leading to greater likelihood of spill periods |
| **Infrastructural change** | | |
| Wear-and-tear | Deterioration of power-generating equipment or dam infrastructure | More frequent outages and maintenance (planned or unplanned) and lower efficiency |
| Capacity additions | Nameplate capacity increases with new unit installations | Planned capacity additions may be associated with an expected decrease in capacity factor if water is constrained |
| **Dam operational change relating to evolving river and reservoir needs** | | |
| Change in reservoir storage rule curves | Lowering target water levels as flood season approaches | Lower reservoir levels lead to reduced hydraulic head and thus less output (MWh) per unit turbined water ($m^3$) |
| Fish passage regulations | Requirements of non-turbined water release to promote fish passage | Decreasing proportion of water passing through turbines for generation |
| General non-power operational changes | Changes to release, such as for navigable waters, water quality control (temperature, dissolved oxygen), and water supply for downstream municipal, industrial, or agricultural needs. | Several possible mechanisms, including increased non-powered spill, and loss of water for generation due to diversions directly from the reservoir |
| **Dam operational change relating to evolving power grid needs** | | |
| Emerging need for ramping capabilities | Hydropower plants increasingly used for their flexibility in balancing variable wind and solar generation | Reduced turbine efficiency of generation with ramping behavior; lower overall powered release |
| Emergence of other must-take technologies | Increased deployment of wind and solar technologies with zero marginal production cost | Reservoir releases are non-powered if grid demands are satisfied by renewables |

than a linear trend over the study period (see "Methods" section for the approach to evaluating the strength of evidence for a shift in mean relative to linear trend). Regions containing multiple dams with an apparent shift in mean CF include New York in the mid-to-late nineties (Fig. 5a), southeastern states post-2000 (Fig. 5b), and Willamette Valley, Oregon, between 2001 and 2005 (Fig. 5c). The shifts are not associated with years of significant generator capacity upgrades or additions. Although further detailed analysis would be required to confirm the causes of these shifts, their presence is more suggestive of operational change than of external environmental drivers. Indeed, six of the seven dams identified in New York (Fig. 5a) and three of the five dams identified in the southeast region (Fig. 5b) had Federal Energy Regulatory Committee (FERC) licenses renewed within six years of observed shifts (shift in annual generation may lag license renewal if new operations are tied to infrastructure investments at the dam). The highlighted dams in the Willamette Valley (Fig. 5c) are Federally-owned and operated (i.e., exempt from FERC licensing). Nonetheless, such projects are also subject to changing operational requirements, and indeed the observed shift at Cougar coincides with the installation of new intake level control infrastructure at that dam to support water quality (temperature) control, which may have been complemented by new water release operations. Further indicating an important role of operational change, we find large long-term declines in CF to be more common among FERC-licensed facilities compared to FERC-exempt facilities (see Supplementary Information, Fig. S4). FERC-licensed facilities account for about 70% of the 610 plants studied, but 92% (12 out of 13) of the plants with the rate of CF change less than −15 PPPD. The median CF trend is also lower across FERC-licensed facilities (−3.02 PPPD, compared to −2.17 PPPD across FERC-except and Federally-owned facilities).

Hydropower dam operational change could also arise as a result of changes in the needs of the power grid. The most obvious and compelling change in energy systems over the last few decades has been the emergence of intermittent, renewable technologies—namely wind and solar, which in the U.S. today contribute generation totals commensurate with the hydropower fleet. These technologies have created a need for real-time balancing resources that can increase output quickly in response to a sharp drop-off in renewable generation, occurring when the sun goes down or when wind speed drops. Hydropower generators are uniquely suited to this role given their ramping capabilities, although providing such flexibility often conflicts with environmental goals[31].

The degree to which these hydropower plants have pivoted toward providing flexibility for intermittent renewables remains undocumented at the national scale. Yet, there are many possible reasons why flexible operations could reduce net annual generation and thus capacity factors. One reason is that flexible operations can impair turbine efficiency. Ref. 32 reports a 4% decrease in simulated generation because of efficiency losses associated with operating for complementarity with wind and solar. Flexible operations could also lead to more spilled energy. Consider the case of a dam that must release some volume of water $Xm^3$ over a day or a week to meet its non-powered obligations. If the plant needs to provide the ability to quickly ramp up generation, it must generate below capacity, and it may not be operated around the optimal efficiency point. If that level of generation requires less than $Xm^3$ water, the difference must be spilled. Indeed, in their case study of two unidentified hydropower plants in the Pacific Northwest, ref. 10 reports that entry into the Western Energy Imbalance Market (EIM) resulted in a loss of net generation, due partly to the EIM requirements for additional flexibility to be available. In practice, there could be a variety of mechanisms through which a change to flexible operations affects annual generation and thus capacity factor. These could present either as slowly evolving operations (as VRE technologies gradually expand their share of grid capacity) or abrupt change (associated with entering a new market setting).

It's important to note that wind and solar technologies have emerged only over the last decade, and thus their influence on trends

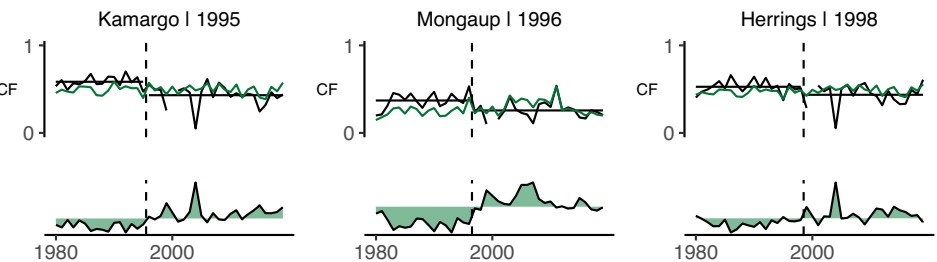

### a  New York, mid−1990s

*Also: Minetto (1997); Blake (1997); Norfolk (1997); Rainbow (1997); Shawmult, ME (1997)*

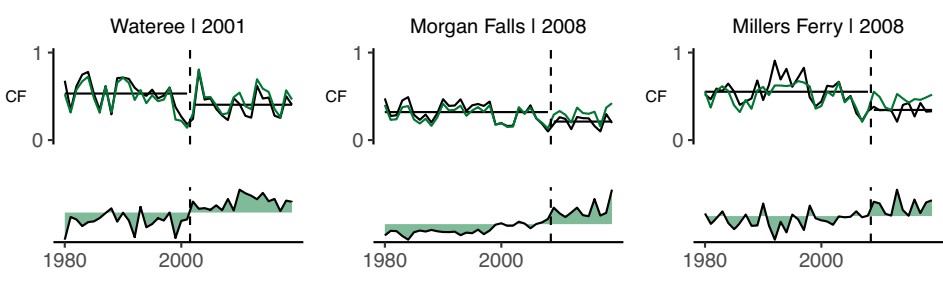

### b  South Carolina / Georgia / Alabama, post−2000

*Also: Buzzard Roost, SC (2001); Flint River, GA (2004)*

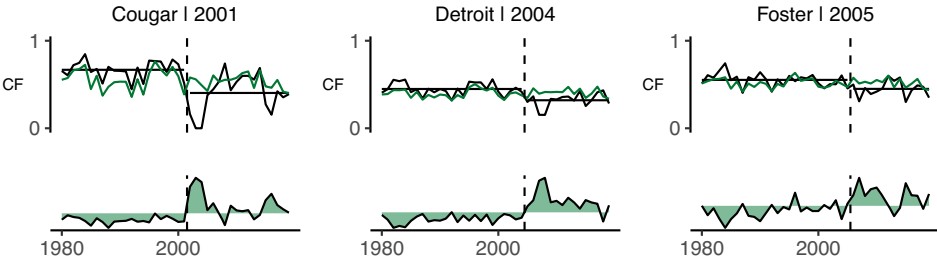

### c  Willamette Valley (Oregon), 2001~2005

**Fig. 5 | Abrupt shifts in reservoir operations are evident in CF when compared to CF simulated with water availability. a** presents three cases in the state of New York. **b** presents three cases in the Southeast. **c** presents three cases in the Willamette Valley, Oregon. The lower time series on each graph gives the residual of the annual CF model (i.e., modeled CF minus observed CF), highlighting a clear point in time when modeled annual CF transitions from underpredicting to overpredicting observed annual CF.

measured over the forty-year period of this study is probably quite limited. Indeed, CF trends observed for the period 1980–2022 are not markedly different from CF trends observed for the period 1980–2009, before the significant uptake of wind and solar in US power grids (see Supplementary Information, Fig. S1). The impact of emergent VRE may present more strongly in the future as the share of grid generation met by wind and solar increases. The extent to which hydropower adapts to this need will depend on operating constraints at individual plants as well as the maturity and economic viability of battery technologies for balancing renewable generation at the grid scale.

In addition to changes in the composition of electricity supply, the last forty years have seen significant growth in electricity demands, as well as possible changes in the seasonality of demand. Seasonal demand change could arise with warming temperatures, which may

lower heating demands in winter and increase cooling electricity demands in summer[23,33]. One could reasonably assume that reduced winter demand could cause a reduction in hydropower generation if coincident with a surplus in supply (i.e., insufficient demand to justify levels of hydropower generation of prior years). However, in reality, hydropower is rarely so dominant in a power grid that its maximal output will fully satisfy demand. Hydropower also has very low marginal production costs, so generation remains profitable even during periods of low demand and low prices. If seasonal hydropower availability exceeds local demand (such as in the Columbia or Colorado River Basins during spring), power is generated and exported[34]. Moreover, hydropower operators are often obligated to release water according to the reservoir management rules designed for non-power objectives, such as complying with flood control curves or maintaining a minimum flow downstream. Since the water must be released,

hydropower plants will tend to generate with as much of it as possible. This contrasts with technologies with high fuel costs or other marginal costs; such technologies have a disincentive to generate when prices are low. Hydropower is the marginal producer in some regions; in those isolated cases, any seasonal shift in load could affect overall output. In general, any historical shift in annual or seasonal electricity demand patterns is unlikely to explain a significant portion of long-term CF decline observed in our study.

### Implications of long-term hydropower generation decline

Despite growing demands for electricity and ongoing efforts to decarbonize the power grid in the United States, this study finds that a significant majority of hydropower plants currently generate less energy annually (per unit installed capacity), on average, than they did in the 20th century. Hydropower plants depend on water, yet our modeling based on streamflow reanalysis data shows that change in water availability cannot fully explain the observed widespread decline in the hydropower generation and capacity factor. Other drivers are needed to explain observed trends in generation and capacity factor, and while external environmental phenomena and infrastructural changes may play a role, we speculate tentatively on the weight of evidence available that changes to dam operations may have been most influential. Confirming the relative importance of different drivers of long-term CF decline will be essential to inform reasonable projections of hydropower generation in the coming decades. The impacts of these drivers on other important hydropower performance metrics, such as dispatchability (ability of plants to dispatch power when needed) and flexibility (ability to quickly ramp up and down output to balance variable loads), constitute an important topic for future research. If the industry has traded total generating performance to meet the evolving needs of downstream communities, aquatic ecology, and power grids, then the CF trends observed here may be expected to continue in line with the nation's environmental goals affecting both river systems and grid composition. The US has significant opportunities for hydropower expansion through new installations[35] and turbine upgrades, which could contribute to a stabilization or possible reversal of the long-term trends we observe. Nonetheless, the possibility of continued long-term change in generation should prompt a rethink on how to represent the hydropower fleet's energy availability in grid planning and electricity portfolio studies.

## Methods

### Annual hydropower capacity factor time series

We analyze capacity factor rather than generation in this study since several plants have seen a change in generating capacity over the last four decades through turbine additions and uprating as well as generator retirements. We study only those plants that have been operational through the entire study period of 1980–2022, focusing on 610 of the 808 plants in the Conterminous United States with nameplate >5 MW (in 2022). These 610 plants account for approximately 90% of overall hydropower capacity in the United States.

To create a time series of annual capacity factors for these hydropower plants, we combine information from different survey datasets provided by the U.S. Energy Information Administration (EIA). Plant nameplate capacities (in MW) are from survey forms 759 (covering years 1980–1986) and 906 (years 1990–2021)[36]. Unit-level status codes provided with each year's survey are used to identify and remove retired generators. The available data leave a gap in nameplate information through the period 1987–1989. Our approach to addressing this gap is a linear interpolation between the 1986 and 1990 nameplates. About 59 plants are associated with a substantial change in reported

nameplate (>10% change) during this period. Most of these changes are associated with capacity increases and upgrades that we can corroborate using various web sources. For 27 of these plants, the data show a substantial reduction in nameplate capacity between 1987 and 1990. Inspection of these cases reveals a discrepancy between form 759 and form 906, with the former (sometimes) including additional installed capacity relating to the pumped storage capability of the plant. We deal with this small number of instances by backfilling the 1990 reported nameplate through the decade 1980–1989. All plant capacities (MW) are converted to implied maximum annual generation (MWh) by multiplying by the number of hours in each year, accounting for additional hours in leap years.

Generation data are from EIA forms 759 (covering years 1980–2000), 906/920 (2001–2007), and 923 (2008–2022)[37]. EIA data collection and quality-checking procedures have been more rigorous since 2001 (based on discussion with EIA). We have no reason to assume that the level of rigor would bias reported generation in one direction or another in the pre-2001 period. Nonetheless, we perform some (limited) verification of the pre-2001 generation totals by comparing them against annual generation totals from the U.S. Army Corps of Engineers, which provides the required data for 13 major plants in the Pacific Northwest, revealing near-identical annual generation totals. With both annual nameplate capacity and annual net generation totals prepared and set to equal units of MWh per year, annual capacity factors are computed for each plant as the total annual generation divided by the maximum generation implied by nameplate capacity in each year.

### Capacity factor trend analysis

Trends for all annual CF time series are computed as the median of slopes between all pairs of data points in the time series. This is known as the Theil–Sen estimator (or Sen's slope)[38], preferred to the ordinary least squares line due to its robustness to outliers. The statistical significance of each plant's trend is established using a bootstrap simulation (10,000 resamples, with replacement). If the autocorrelation coefficient of the annual CF time series at lag 1 year exceeds 0.3, a block size of 3 years is applied to preserve autocorrelation in the resamples. The $p$-value is equal to the proportion of randomly generated samples with a trend of greater magnitude than the trend in the original annual time series. To illustrate, a $p$-value of 0.2 in a plant with an observed decreasing trend of 1.5 percentage points (pp) per year indicates that 20% of randomly generated samples resulted in a decreasing trend greater than 1.5 pp per year. Throughout the study, we use a $p$-value of 0.05 as a threshold for statistical significance in the trend.

Trend analysis is performed on the full time series (1980–2022) as well as on a reduced time series (1980–2019) that is aligned with the available streamflow data. The full time series is used to report observed CF trends (i.e., Fig. 1), while the CF trends computed for 1980–2019 are used for like-to-like comparison against CF trends simulated using annual water availability. A comparison of trends computed over these two time periods is given in Fig. S1 of Supplemental Information, showing minimal differences between the two.

### Water availability data

Water availability data are from DayflowV2[38,39]—a 40-year (1980–2019) historical streamflow reconstruction that provides hourly, daily, and monthly-resolution data for approximately 2.7 million NHDPlusV2 stream reaches in CONUS. In this study, we use Dayflow forced with Analysis of Record for Calibration (AORC) reanalysis precipitation[40,41]. Total runoff (i.e., baseflow and surface runoff) simulated with the Variable Infiltration Capacity (VIC)[42] model is routed through the NHDPlusV2 network using the Routing Application for

Parallel computation of Discharge (RAPID)[43,44] to produce natural streamflow. Streamflow observations derived from approximately 3000 USGS streamflow gauges are then assimilated to provide a non-natural ("Assimilated") setting that serves as a historical reanalysis streamflow dataset to represent actual hydrological changes in this study. Further details on model calibration and streamflow performance across CONUS at over 7500 USGS stream gauges are provided in ref. 38.

Dam assignment to stream reaches is achieved using the Hydropower Infrastructure – Lakes, Reservoirs, and Rivers (HILARRI) database (version 2)[45]. This dataset provides the NHDPlusv2 stream reach (identified using the "COMID") associated with individual dams and reservoirs. Of the 610 plants analyzed for CF trend in this study, 577 are associated with an NHDPlusV2 stream reach in HILARRIv2. With the stream reach identified, we extract from DayflowV2 the annual streamflow totals for each plant and then simulate the plant's CF as a function of water availability using the modeling approach outlined in the following section. The final set of modeled plants numbers 362 due to the removal of plants with flow data that are significantly biased or fail to capture the interannual variability of annual CF observations. These traits indicate possible misallocation of the stream reach or inaccurate hydrologic processes representation in Dayflow—which can occur for a variety of reasons, including uncertainties in inputs, models, and observations. Removal of these data reduces the number of plants studied but strengthens confidence in overall results on the effects of water availability on the CF trend. One notable effect of this final filter is the removal of many dams in California. This is partially a result of dense conduit and water transfer networks that are hard to accurately map in the NHDPlusV2 network and whose effects are not captured well in CONUS-scale streamflow routing models.

## Capacity factor model based on annual water availability

Although inappropriate for modeling sub-annual generating patterns from hydropower plants, annual hydropower generation has been modeled quite successfully using linear models in the form $E = \alpha + \beta \times Q$, where $E$ is total annual energy generated, and $Q$ is a variable representing water availability, which could be total annual inflow to the reservoir, total upstream runoff, or even total upstream water-year precipitation[46–48]. The main problem with a linear model in this setting is the overestimation of generation in extreme years, both wet and dry. For a wet year, expected generation should level off to account for the capacity of the plant and increased non-powered spill; for dry years, reduced hydraulic head, possible deadpool conditions, and reservoir contingencies to conserve water combine to reduce generation relative to water available. Moreover, a linear model is flexible enough to accommodate any level of bias in the hydrological input, leading to its intercept, $\alpha$, often serving as a bias adjustment factor and thus losing any physical interpretability.

To address these issues, we adopt an alternative method that better captures generation in extreme water conditions while also aiding in the identification and removal of significantly biased water availability data. The approach is analogous to the Budyko framework applied in the field of catchment hydrology[49]. The Budyko model estimates long-term average evaporation, $Ev$, as a function of long-term average potential evaporation, $Ev_p$, in the form $\frac{Ev}{Pr} = f\left(\frac{Ev_p}{Pr}\right)$, where Pr is long-term average precipitation. We adapt this to $\frac{E}{E_{max}} = f\left(\frac{E_p}{E_{max}}\right)$, where $E$ is the actual annual energy generated by a hydropower plant, $E_{max}$ is the annual energy generation maximum implied by plant nameplate capacity (meaning $\frac{E}{E_{max}}$ is the plant's capacity factor), and $E_p$ is the total potential energy available in the annual streamflow at the plant. We term $\frac{E_p}{E_{max}}$ the "capacity potential." Since the predictand

(capacity factor) and predictor (capacity potential) share the same denominator (nameplate capacity), and since actual generation cannot exceed the potential energy in the annual flow, there is a 1:1 slope from the origin that separates feasible from non-feasible capacity factors for given water available. We term this the "energy limit." There is also an upper limit of capacity factor equal to 1 (the "capacity limit"). Conveniently, these two physical limits on generation bound the asymptotes of the flexible single-parameter curve (Eq. 1):

$$CF = 1 + \varphi - (1 + \varphi^{\tau})^{1/\tau} \qquad (1)$$

where $CF$ is the annual capacity factor ($\frac{E}{E_{max}}$), $\varphi$ is the annual capacity potential ($\frac{E_p}{E_{max}}$), and $\tau$ is the single parameter that defines the curve's position within the feasible solution space (this curve is from ref. 50). To compute the capacity potential for each year (the input to this model), annual total water availability (i.e., total streamflow at the dam, taken from the streamflow reanalysis data, Dayflow) at each dam is converted to potential energy, which is then divided by maximum energy implied by the plant's nameplate ($E_{max}$):

$$\varphi = \frac{\rho \times g \times h \times Q \times 2.78 \times 10^{-10}}{E_{max}} \qquad (2)$$

where $\rho$ is the density of water (1000 kg/m³), $g$ is the acceleration due to gravity (9.81 m/s²), $h$ is the hydraulic head (constant) of the plant (m), and $Q$ is the total volume of water in streamflow (m³). The constant $2.78 \times 10e^{-10}$ converts potential energy from Joules to MWh to align with $E_{max}$. The hydraulic head is from the Hydropower Energy Storage Capacity Dataset version 2[51].

Despite its simplicity, this single-parameter CF model provides the nonlinearity needed to capture typical behavior in wet and dry years. Moreover, the model will fail to fit a curve to heavily biased inputs (due to its physical limits) and can therefore also be used to easily identify instances where input data (water availability, used to derive potential energy) are unsuitable for analysis. This filter leads to us modeling CF for only 362 plants rather than the 552 hydropower plants with streamflow data available. A two-parameter version of the model allows for the condition of zero generation in years of non-zero water availability. The physical justification for this additional parameter is that generation can be curtailed dramatically with low flows as a result of drought contingency measures. The two-parameter model is:

$$CF = 1 + \varphi - (\gamma + \varphi^{\tau})^{1/\tau} \qquad (3)$$

where $\gamma$ is the second parameter controlling the orientation of the curve, constrained such that $\gamma \geq 1$ (since $\gamma < 1$ would lead to instances of generation exceeding potential energy in the water available). Graphical representations of these models are given in Fig. S2 and example model results for a run-of-river and a storage dam are given in Fig. S3. We use the two-parameter model (Eq. 3) in our study, although for most run-of-river and small storage cases the second parameter is not required. Parameters are fitted using the quasi-Newton method of ref. 52. Extensive leave-one-year-out cross-validation conducted across hundreds of dams demonstrates that this model improves the capture of CF during wet and dry years relative to a linear model with the same number of degrees of freedom (i.e., two parameters) (Fig. S4).

## Proportion of CF attributable to water availability

We fit the CF model (Eq. 3) for all 362 plants with adequate streamflow (after removing cases of obvious bias as revealed by the computation of potential energy in the flow) using the entire 1980–2019 streamflow reanalysis period. Since the CF model is forced only with water availability, changes in CF that arise due to factors other than water

availability emerge as a trend in the residuals between observed CF and modeled CF. The proportion of any negative CF trend explained by water availability is simply the slope of the modeled annual CF (1980–2019) divided by the slope of the observed annual CF for the same period. For each plant, annual CF is modeled using both the gauge-assimilated streamflow and the natural streamflow. Cases for which CF trend can be explained by CF modeled with gauge-assimilated streamflow but not by CF modeled with natural stream-flow are those that we identify as being driven by influences on water availability other than climate.

To assess whether some cases with significant CF decline not predominantly driven by water availability are more likely to have arisen due to a shift in the mean rather than a linear trend, we adopt a similar bootstrapping procedure as used to evaluate statistical significance in the CF trend. This analysis is conducted on model residuals (modeled CF – observed CF) to remove the influence of water availability. We first identify the split year that results in the largest difference of means of the annual model residual series. We then perform the bootstrap with 10,000 resamples (as in the trend analysis above) and determine both the slope and the difference in means in each random sample. The bootstrap results are then used to evaluate relative likelihood (p-values) for both the observed difference in means and observed slope. A similar approach is adopted to test for capacity additions being a driver of CF trend. In this case, we simply select the year of largest capacity addition and test whether the shift in mean of the residual presents more statistical significance than the linear trend.

### Evidence for flow regime change driving CF trend
Daily flow data are compiled for 107 plants which show both statistically significant CF decline and for which trend in CF simulated with annual water availability explains less than 60% of this decline. Based on these data, we compute the following metrics: annual 90th percentile of daily flow, annual maxima of daily flow, annual maxima of 7-day flow moving average, annual maxima of 30-day flow moving average, and proportion of annual flow arriving in the wettest month.

## Data availability
All data used in this study are publicly available and may be used to reproduce all results, data, and graphics using a comprehensive data pipeline coded in the R {targets} framework (https://code.ornl.gov/turnersw/hydro-cf-trends/). Hydropower plant locations on NHDv2 river network are from HILARRI. Hydropower plant attributes are from the Existing Hydropower Assets (EHA) database (https://doi.org/10.21951/EHA_FY2022/1865282) and the Hydropower Energy Storage Capacity Dataset (HESC) (https://doi.org/10.21951/HESC/1972462). All hydropower data (generation totals and nameplate capacities) are from Energy Information Administration, forms 923, 906, 920, 860, and 759 (https://www.eia.gov/electricity/data.php). Reanalysis streamflow data are from Dayflow Version 2 (https://data.ccs.ornl.gov/ui/doi/464). A consolidated input dataset has been deposited in the HydroSource database along with results generated in this study[53] (https://doi.org/10.21951/hydro_trends/2349418).

## Code availability
This study may be reproduced in its entirety from a single, comprehensive data pipeline coded in the R {targets} framework. The reproducible data pipeline, which includes the reproduction of all graphics presented in this article, is stored on the following repository: https://code.ornl.gov/turnersw/hydro-cf-trends/-/releases/1.0.0.

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

## Acknowledgements

This research is supported by the US Department of Energy (DOE) Water Power Technologies Office as a part of the SECURE Water Act Section 9505 Assessment. We thank Daniel Broman for his improvements to HILARRIv2 stream reach identification for a selection of dams, and we thank Rocío Uría-Martínez and Megan Johnson for their insights on U.S. hydropower data and industry trends.

## Author contributions

S.T. and S.C.K. conceived the study. S.T., S.C.K., D.S., G.G., and C.H. formulated the analysis. S.T. led data collection, analysis, and manuscript preparation. G.G. and S.C.K. prepared streamflow data for natural and gauge-assimilated settings. C.H. and D.S. prepared hydropower and dam data, including stream reach linkage to plants. S.T. created the annual capacity factor streamflow model. S.T. developed the reproducible data pipeline and generated visuals. All authors contributed to manuscript preparation.

## Competing interests

The authors declare no competing interests.
