## [Peer Review File · Nature Communications]

Hydropower capacity factors trending down in the United StatesReviewers' Comments:

Reviewer #1:

Remarks to the Author:

Thank you for the opportunity to review. I'm glad to see more attention to the hydropower fleet. I think this is a worthwhile question/topic and my main feedback is that I think the authors make a number of claims that are not super well supported regarding what drivers could be, but which could be fairly readily tested with available data. My biggest concern is that the paper does not address temporal mismatches related to seasonal water availability as a potential water availability-driven influence on the (fairly small) observed CF declines.

L16: is "utilization" the right term here? I get what you mean but I think there are differences in how people read this in a technical context — e.g., I might think of utilization as "running the plant while it's available" vs availability as "how frequently the plant is actually available." not a huge deal but I might use something more specific like "output" or just "capacity factor" here.

L20: "sacrificed" has a particular values valence — I might use a term like "exchanged" or something more neutral. relatedly: any chance the decline is due to age or similar?

Abstract general — suggest stating your time period of analysis. Over what period is that negative trend, and how consistent is it (e.g., is it monotonic between 2010-2022?)

L33: I don't think this is true. As with other renewables, we expect them to run when fuel (i.e., water) is available and when there's demand, both of which limit operations

L51: I know what you mean but it wasn't immediately clear that you're making a point about the denominator increasing

L57: what we observe is that fossil fuels are backstopping, but that's not "necessitated" per se. suggest something like "which has so far lead to replacement generation from.."

L74: suggest including some stats here about how much of the fleet is covered

L75: cf instead of generation — did you also check generation to see if this is potentially a result of e.g. insufficient demand to run more? I agree that the CF is important given capacity changes but I think the argument that plants are running less is weaker without a confirmation that the gen is also lower over time.

L83: that's a pretty small impact given the way the loss is characterized in the abstract that makes this decline sound very significant

L85: again to the question of whether this is partly due to demand patterns, did you confirm that the water is available at times when it is actually useful? e.g., high winter water availability might not lead to much use given lower winter demand so far, but it's also useful to flag seasonality mismatches because of dynamics like an expectation that we'll move to winter peaking as heating electrifies (which could make these dams extremely valuable).

L86: what does "climate-driven" mean here? is that explicitly the impact of climate change, or a regional thing?

L90-93: I think this is a much more measured way of presenting the results than is in the abstract

L97: I think it would be helpful to discuss the populations of dams with + vs - CF in terms of total capacity and/or generation, in addition to number — e.g., it's important to know whether all the little dams are seeing declining CF while the bigger ones are mostly increasing when judging the influence of different drivers. I see you get into this a bit later but stating right up front "482 dams (X% of capacity, Y% of generation) would be valuable I think.

L126: might be useful to note that the PNW dominates overall US hydro production, so we'd expect the trend to be similar for the country as a whole

Fig2: what's the time interval here (e.g., annual)? (suggest adding this to all relevant figures)

L182: did you look at this from a seasonal perspective? re above I would expect to see generation be limited if there's suddenly a lot more water during lower demand periods and a lot less water during higher demand periods.

L186: just flagging the paper inconsistently uses gauge vs gage (I'd probably go with gage)

Table 1: on the "power grid needs" side I think not accounting for temporal mismatch between supply availability and demand potentially overlooks a major driver here

L242-245: I think it would not be that hard to check this given that EIA generation data are issued monthly. even investigating a few of the big dams and whether their seasonal patterns have changed would be pretty important for claims I think.

L263: again here I think you can check this in your data. what are the generation trends? relatedly, EIA has age data and it wouldn't be too hard to check whether there's a big difference in CF trends for older dams (though sometimes major overhauls are tricky to spot)

L288: I think it's tricky to make the claim that this is "perhaps the most likely cause" based on your L298 and the fact that there are some checkable things about seasonality, plant age, generation trends, etc that would tell you how significant those factors are. The hydro literature tends to implicitly demonize environmental protections as a driver of reduced hydropower outputs and I think it's important to be really careful not to fall into that without evidence, even if unintentionally. As written, the paper seems to pretty heavily imply that it's environmental rules and other renewables that are driving these issues, but I don't see a ton of evidence for that implication.

L303: maybe, but I think only if sustained: I'd actually expect mechanical issues to cause a pretty abrupt change.

L330: possibly? but is that super relevant in places with a lot of hydropower? I'm not convinced this is a huge issue in the PNW, for example, though issues like increased summer load due to population growth + increased need for cooling, which could throw off seasonal matches between hydro availability and power demand, could be important. again I think this is pretty verifiable (check to see if big increases in wind/solar in places with a lot of hydro correlate with downward CF trends) and probably worth doing.

Conclusion/throughout: I think the paper hits the point about downward CF trends a little hard for the size of the CF decline. I'd like to see a nod to the decline in generation in percentage terms as well (e.g., if the CF is declining by 0.5%/year but gen is declining by 0.05%/year, that's relevant, and a consistent statement of the size of the decline when it's brought up.

L382: interesting closing given that I feel one of the main args of the paper is that the decline is reversible since it's not mostly based on resource constraints?

Reviewer #2:

Remarks to the Author:

Dear Authors and Editors,

This is the first time in my long history of reviewing to present my comments without reading the full paper but in the present case, I feel strongly that I don't need to read it any further.

I think the paper is fundamentally wrong in lumping together the capacity factors of different energy sources. The authors themselves acknowledges that peaking technology gas turbines has a meager 13% capacity factor, end yet they compare the hydropower capacity factors to other forms of energy such as solar or nuclear and discuss its decline in hydropower generation as if it was a negative trend that would limit the value of hydropower in the future.

The paper itself recognizes that the change in capacity factor of hydropower is not necessarily tied to the availability of water resources and noted that there are instances when the water resources increased and yet the capacity factor decreased due to the hydropower industry "sacrificing" portion of generation to support environmental objectives and more importantly "serving emerging grid needs".

I think the authors should seriously revise their paper and reconsider what the "declining" capacity factor is meant to measure. Based on the title itself, one would think that decline is a sing of another "looming" climate change driven disaster, but when it comes to the capacity factor of hydro power vs. other forms of renewable energy sources there is a big difference.

Hydropower is often used for load balancing since this is the fastest means on the grid to add rapidly generation capacity. The "serving emerging grid needs" is likely the result of solar and wind flooding the grid necessitating hydropower intermittently kicking in as solar and wind abruptly drop out and turn off rapidly when solar or wind suddenly ramp up. Just like the low capacity factor of peak generating gas turbine is the result of their ability to provide power when needed and stay idle when the demand decreases or the supply ramps up, the same is true for hydropower, so the decline have less or nothing to do with the availability of the resources, but utilizing its flexibility in providing power as needed. Low capacity factor of solar and wind is a measure of their failure to serve their purpose.

Without recognizing these differences and evaluating whether the lower capacity factor arose from a necessity of load balancing or resulted from diminishing water resources the discussion of trends in capacity factors is either pointless or severely misleading.

I don't think this paper can be published without clarifying these differences, but I will be happy to read it carefully read it and provide constructive suggestions, once the authors addressed this fundamental comment.

Balázs M. Fekete
Department of Civil Engineering
City College of New York
140 Convent Ave. New York, NY 10031

Reviewer #3:

Remarks to the Author:

What are the noteworthy results?

This work provides strong, well-supported evidence of a long-term decreasing trend in the capacity factor of the U.S. hydropower fleet and of that decrease being only partially attributable to changes in water availability.

Will the work be of significance to the fields and related fields? How does it compare to the established literature? If the work is not original, please provide relevant references.

To my knowledge, this multi-decadal, plant-level analysis of U.S. hydropower capacity factor using statistical methods to perform attribution to water availability is novel.

Improved understanding of capacity factor trends in the U.S. hydropower fleet and their drivers is of significance not only to the field of hydrology but to electricity grid models which sometimes make overly simplified assumptions about hydropower generation.

Does the work support the conclusions and claims or is additional evidence needed?

The conclusions are properly supported with a combination of (a) thorough data analysis involving linkage of capacity, generation, and streamflow datasets and (b) simulation techniques and statistical tests. All the datasets used (EIA capacity and generation data, USGS' NHDPlus, ORNL's Dayflow) are publicly available with detailed methods to understand their scope and limitations. The use of natural and gauge-assimilated streamflow data to simulate capacity factor provides extra insight in that it allows separating changes in water availability due to climate versus other reasons such as reallocation of water to non-power purposes. The discussion of potential other drivers for the capacity factor declines is comprehensive and supported by some specific examples.

I commend the authors for the effort to produce high-quality visuals that combine maps and plots. These are a great way to summarize complex results where both spatial and time-series dimensions are of interest.

Are there any flaws in the data analysis, interpretation and conclusions? Do these prohibit publication

or require revision?

I have some suggestions for minor revisions but would overall recommend this manuscript for publication.

- Line 33: "hydropower plants operate with an annual CF of just 20-50% varying significantly by dam and year of operation". How comprehensive is the dataset used in the paper from which this statement is referenced? In my own analysis of capacity factor data combining EIA Form 923 and EIA Form 860, I have found instances of CFs higher than 50%. Often, they are small run-of-river plants which might not be part of the dataset in the referenced paper.
- Line 48: "Today in the United States, hydropower contributes approximately 7% of electricity generation and 38% of renewable generation." The percentage of renewable electricity seems too high. I've seen numbers under 30% in recent years.
- I suggest stating the percentage of U.S. capacity captured by the various subsets used in different stages of the analysis. Specifying the capacity percentage covered helps assess whether all findings can be interpreted as representative of the entire fleet. For instance, the 610 hydropower plants for which historical capacity factor data are available represent 90% of U.S. capacity. What percentage of capacity do the 362 plants for which gauge-assimilated streamflow data are available represent?
- The initial discussion of observed capacity factor trends refers to differences by nameplate capacity ("declining capacity factor trends are particularly common at the largest plants by nameplate capacity") and region ("declining hydropower CF appears throughout the nation but is most prevalent in the West"). I suggest exploring any noteworthy differences in trends along two other dimensions:
 - 1) Federal vs nonfederal (FERC-licensed) fleet
 - 2) Peaking versus run-of-river plants. I expect that decreased CF due to dam operational change relating to evolving environmental mitigation requirements might be more pronounced for peaking plants (e.g., ramping restrictions and restrictions in reservoir fluctuations added during relicensing).
- In the discussion of operational changes as likely important drivers of declining CF, it might be worth to mention the Electric Consumers Protection Act of 1986. It required FERC to balance power and non-power values when issuing hydropower licenses. There is some evidence that projects relicensed post-ECPA, especially peaking projects, lost capacity during relicensing (due to increased weight given to environmental objectives). In their identification of one-time shifts in capacity factor, the authors state that the shifts do not neatly align with dam relicensing years. I suggest checking if the one-time shifts are more common for relicenses issued post-ECPA and if one-time shifts typically happen post-relicensing. It might take a few years for the licensees to implement the changes required in a relicense (especially if they involve capacity investments such as fish passage facilities) so operation shifts lagging relicensing a few years might still be driven by relicense conditions.
- An additional potential driver of abrupt changes in operations are changes in the market setting/structure in which the hydropower plant is operating. For instance, the Datta et al. (2022) paper referenced goes over changes in the operations of two hydropower plants after joining the Western Energy Imbalance Market. This is presented under the discussion of changes in operation due the needs of the power grid. However, evolving grid needs would likely show up gradually in the data while joining the WEIM (or another organized market with different bidding requirements/rules) can cause more abrupt changes.
- In the conclusions or elsewhere in the discussion, I suggest mentioning that capacity factor is not the only metric that matters to evaluate the performance of a hydropower plant (regarding its power purpose). The availability of capacity at the times when it is most valuable and the flexibility to adjust output up or down with short notice are also important. Different drivers of capacity factor decline would affect dispatchability and flexibility differently which is another reason why further work to disentangle the relative importance of the various drivers proposed would be worthwhile.

Is the methodology sound? Does the work meet the expected standards in your field?

I'm not a hydrologist and cannot comment on whether a different hydrology model would be better suited to analyze the data considered in this study. However, the authors provide a well-reasoned justification for their chosen methodology and it has been applied rigorously. The authors demonstrate improvements in fit for the chosen "new CF" model versus a simpler linear model of capacity factor.

Is there enough detail provided in the methods for the work to be reproduced?

Yes, the authors provide a detailed description of the data used and equations estimated. Additionally, they state that they will make a repository available that will allow replication of the analysis upon publication.

Reviewer 1

R1-Summary: *Thank you for the opportunity to review. I'm glad to see more attention to the hydropower fleet. I think this is a worthwhile question/topic and my main feedback is that I think the authors make a number of claims that are not super well supported regarding what drivers could be, but which could be fairly readily tested with available data. My biggest concern is that the paper does not address temporal mismatches related to seasonal water availability as a potential water availability-driven influence on the (fairly small) observed CF declines.*

Authors' response: Thank you for your constructive comments and careful attention to detail. We appreciate your time and effort in conducting this review. We agree with almost all your comments, including the suggestion to provide further analysis on seasonal hydrological change as a potential driver of CF trend. The comment that observed trends are “fairly small” may be due to some misinterpretation of a particular number reported in our initial manuscript. Our analysis finds that 80% of hydropower plants have experienced long-term decline in CF, and that the aggregated effect of these CF declines translates into a fleetwide, cumulative generation decrease of 23% since 1980 under static capacity, or a 13% loss when factoring in capacity upgrades. We hope our clarifications on rates of CF decline, as well as additional analyses on trends in net generation, help demonstrate to you the significance of the observed CF declines.

For your convenience in finding our revisions in the revised manuscript, line numbers referenced below refer to the document *with* tracked changes. For brevity of reply, we have grouped some of your comments into general common threads (demand mismatch, interpretation of trend, etc.).

R1-1: *L16: is “utilization” the right term here? I get what you mean but I think there are differences in how people read this in a technical context — e.g., I might think of utilization as “running the plant while it’s available” vs availability as “how frequently the plant is actually available.” Not a huge deal but I might use something more specific like “output” or just “capacity factor” here.*

R1-2: *L20: “sacrificed” has a particular values valence — I might use a term like “exchanged” or something more neutral. Relatedly: any chance the decline is due to age or similar?*

Authors' response: We agree with both suggestions on wording. Please see response to comment **R1-20** below for analysis of plant age.

Actions taken: We have changed “utilization” to “generation” and removed the passage containing the word “sacrificed.” (LINES 17 and 31).

R1-3: *Abstract general — suggest stating your time period of analysis. Over what period is that negative trend, and how consistent is it (e.g., is it monotonic between 2010-2022?)*

Authors' response: We agree that both the time period and results of trend significance testing should be present in abstract.

Given the use of annual generation totals, and the very large interannual variability present in these time series data, analysis of trends over shorter time periods rarely reveals statistically significant patterns. Nonetheless, we have now repeated our analysis over the period 1980–2009 (30-year period), providing further insight into CF trend over the period prior to significant renewables and before major drought events affecting western regions in recent years (e.g., California 2013–2015; 2019–2021). This new analysis lends further evidence to the robustness of trends reported.

Actions taken: We have added the time period of study and trend significance testing results to the abstract (LINES 11, 18). We have added new analysis covering the period 1980-2009 (LINES 188-190, 495-499, Supplementary Information Figure S1).

R1-4: *L51: I know what you mean but it wasn't immediately clear that you're making a point about the denominator increasing.*

Authors' response: We agree this was unclear.

Actions taken: Changed to “The prospect of long-term change in *annual* hydropower CF generation matters” (LINE 81).

R1-6: *L57: what we observe is that fossil fuels are backstopping, but that's not “necessitated” per se. suggest something like “which has so far lead to replacement generation from.”*

Authors' response: Right—it could in theory be handled with load curtailments or other resources (if available). We agree with the suggested wording change.

Actions taken: We have amended the wording as suggested (LINE 91).

R1-7: *L74: suggest including some stats here about how much of the fleet is covered.*

Authors' response: Agreed.

Actions taken: We have added that the analysis covers 75% of plants > 5MW and 87% of total US installed conventional hydropower capacity (LINES 117–119).

R1-8: L75: *cf instead of generation — did you also check generation to see if this is potentially a result of e.g. insufficient demand to run more? I agree that the CF is important given capacity changes but I think the argument that plants are running less is weaker without a confirmation that the gen is also lower over time.*

Authors' response: Thank you. This suggestion to analyze generation directly has led to an important addition to the paper.

A critical clarification first. Capacity factor is not proportion of time running; it is actual generation divided by maximum possible generation implied by the nameplate (consistent with most formal definitions, including EIA: https://www.eia.gov/tools/glossary/index.php?id=Capacity_factor). We are therefore not necessarily showing that plants with declining CF are running less. They are producing less energy relative to their capacity, which could be the result of running less *or* having lower output while running (or both). The latter could result from lower conversion of flow to energy (impaired turbine efficiency or lower average head levels, for example) or reduced flow through turbines and thus lower output while running. For a plant that has not changed its nameplate capacity (about half of plants studied), the CF analysis produces the exact same relative trend as would an analysis of generation directly.

Nonetheless, we agree that a complementary analysis of impacts on total generation would add valuable new insight, since this analysis would reveal the extent to which capacity upgrades have counteracted the CF decline observed. We have now performed this analysis. A total of 342 plants out of the 610 plants analyzed have had capacity added in over the last 43 years. While CF has declined at 80% of plants, these capacity additions have led to actual generation declining at approximately 70% of plants. In other words, even though most plants have added some capacity during the last 43 years, only 30% of plants have increased generation over this time period. Moreover, of the 298 plants with declining CF and which have had capacity added at some point over the last 43 years, 224 plants (75%) experienced decline in generation despite having upgraded their capacity.

With regards to insufficient demand, please see our detailed response to group of comments **R1-11**, **R1-12**, **R1-13**, and **R1-14** below.

Actions taken: We have added new results and discussion on the effect of capacity additions (LINES 208-230), including a new figure in Supplementary Information comparing actual CF trend to CF trend assuming fixed capacity (Figure S2).

Comments questioning the magnitude of impact

R1-9: *L83: that's a pretty small impact given the way the loss is characterized in the abstract that makes this decline sound very significant.*

R1-10: *Conclusion/throughout: I think the paper hits the point about downward CF trends a little hard for the size of the CF decline. I'd like to see a nod to the decline in generation in percentage terms as well (e.g., if the CF is declining by 0.5%/year but gen is declining by 0.05%/year, that's relevant, and a consistent statement of the size of the decline when it's brought up..*

Authors' response: Thank you for these comments, which signal to us that some of our wording on the magnitude of the overall impact may have been unclear.

Please note that the paper does not present *any* results as % change in CF. We use PPPD (percentage points per decade) throughout. For example, a change in CF from, say, 0.32 to 0.24 over a decade would be a decrease of 8 percentage points. Such a decrease in CF implies a **25% decrease** in generation over that decade at a plant with no capacity upgrades. We report it as -8 PPPD.

We find that the median trend in generation across 610 plants is -2.6 PPPD. So more than half of hydropower plants have lost more than 11 percentage points in CF since 1980 (2.6 multiplied by 4.3 decades). Given the very large interannual variability of hydropower generation, one cannot find statistically significance in trends unless the trend is also large in magnitude. We show that two-thirds of declining trends are significant at the 5% level. The aggregated, fleetwide impact implied by these CF changes is a **cumulative loss of about 23% generation since 1980** (or about a 13% loss once capacity upgrades are accounted for). This is a nontrivial magnitude of decline for the hydropower industry and for researchers engaged in long-term power planning studies.

The abstract does not report any result that is not demonstrated in the main text of the study. Significant, widespread reduction in CF (and plant output) is the key result of the study, summarized faithfully in the abstract. The use of Hoover dam output helps put the result in context (this is a facility most readers are familiar with), although we agree that this should be accompanied with the % impact in the abstract (since many readers may assume Hoover accounts for much more than 1% of overall fleetwide generation).

The 0.5%/year number (inferred in your above comment to be CF change) was in fact already converted to implied impact on generation at the US scale:

“...overall change in capacity factors, across all plants studied, **implies a net loss of 1.82 TWh per year** across the hydropower fleet”.

This sentence was perhaps worded rather confusingly, and can give the false impression that the fleetwide, long-term impact we see is a loss of about 0.5% of generation. Rather, we were stating that the *annual* rate of loss is equal to about half a percent of contemporary average annual

output. Since this is a confusing way to summarize the aggregate impact, we have now instead opted to tabulate the aggregate CF trend along with implied generation trend (i.e., assuming no capacity upgrade), and actual generation in terms of cumulative impact since 1980. Note that a small correction in our computation has led to a revision of the rate of change in annual generation to be -1.5 TWh/year.

Actions taken: To further elaborate the meaning and import of the CF changes reported in our analysis, we have added a new summary table that reports three key metrics at CONUS and regional scales: (1) overall change in regional CF (in PPPD), (2) change in generation implied by CF change (i.e., computed assuming no change in nameplate since 1980), and (3) change in actual generation, accounting for changes in nameplate since 1980. The comparison between (2) and (3) reveals the extent to which capacity upgrades have counteracted declining CF (Table 1 – LINES 225–231).

Comments on possible effects of seasonal demand patterns on hydropower generation

R1-11: *L33: I don't think this is true. As with other renewables, we expect them to run when fuel (i.e., water) is available **and when there's demand**, both of which limit operations.*

R1-12: *L85: again to the question of whether this is partly due to **demand patterns**, did you confirm that the water is available at **times when it is actually useful**? E.g., high winter water availability might not lead to much use given lower winter demand so far, but it's also useful to flag seasonality mismatches because of dynamics like an expectation that we'll move to winter peaking as heating electrifies (which could make these dams extremely valuable).*

R1-13: *L182: did you look at this from a seasonal perspective? Re above I would expect to see generation be limited if there's suddenly a lot **more water during lower demand periods and a lot less water during higher demand periods**.*

R1-14: *Table 1: on the "power grid needs" side I think not accounting for **temporal mismatch between supply availability and demand** potentially overlooks a major driver here.*

Authors' response: The above comments suggest a link between seasonal electricity demand and total output from hydropower, implying that the former affects the latter. While such a link is intuitive, the reality is that output from U.S. hydroelectric power plants is insensitive to typical demand fluctuations at seasonal timescales. Hydropower generation often follows load (or price) at sub-daily (sometimes sub-weekly) timescales. But at monthly, seasonal, or annual timescales, total generation follows water availability, not demand. Some explanation and demonstration of this phenomenon follow.

In general, hydropower is rarely so dominant in a power grid that its maximal output will fully satisfy demand. If hydropower does satisfy local demand, power is generated and exported. In fact, excess hydropower was a major driver of grid transmission in the United States. Here is a passage from the recently published FERC Primer on Energy Markets: "*Large federally owned*

dams on the Columbia and Colorado river systems generate power from the Spring runoff of melting mountain snow. When the reservoirs are full and hydroelectric plants are generating plentiful amounts of power, there is not enough local demand to use the supply. Since hydropower was cheaper than any alternative, long distance transmission lines were built to deliver the excess power from the Northwest and Southwest to load centers in California” (FERC, 2023).

Hydropower has almost no marginal production costs, so generation remains profitable even during periods of low demand and low prices. Moreover, hydropower operators are often obligated to release water according to the reservoir management rules designed for non-power objectives, such as complying with flood control curves, or maintaining a minimum flow downstream. Since the water must be released, hydropower plants will tend to generate with as much of it as possible, irrespective of electricity price. They may try to shape generation to balance short-term load and maximize revenues, but the overall generation within a month is not a function of total demand. This contrasts with technologies with high fuel costs or other marginal costs; such technologies have a disincentive to generate when prices are low. Fuel not used during low price periods can be stored for later use; it’s not lost to spill as with hydro. Even in the Northwest, where hydropower represents a very large proportion of generating capacity, the low demand season (spring) coincides with the highest output from hydropower units due to high flows driven by snowmelt. If the potential generation exceeds local demand, hydropower will be produced and exported elsewhere in the Western grid.

Since this issue was highlighted as a main concern, we have decided to back up our reasoning with some further analysis. Below we show data for the Pacific Northwest (PNW) to demonstrate the disconnect between electricity demand and hydropower output—even in this area where hydropower dominates electricity supply.

Figure 1 – Does demand drive hydropower output in the PNW? Analysis shows no correlation between PNW hydropower output and both Western grid demand (left panel) and local PWN demand (data from EIA monthly electricity dashboard). Highest output from hydropower occurs in spring, when demand is low. Often the generation from hydropower exceeds the local demand, indicating export to elsewhere in the grid.

Please note that the above commentary does not imply that seasonal shifting of water availability and load are unimportant from a power grid reliability perspective. In fact, our lead author published on this very subject in *Nature Communications* (Turner *et al.*, 2019), highlighting the importance of compounding changes to both seasonal hydropower availability and load under climate change. Here we are simply saying that any trend in seasonal demand patterns would not have influenced the annual output at hydropower plants. Even if seasonal shifts were present in the demand data, we would not attribute any CF trend to those shifts.

Federal Energy Regulatory Commission, 2023. Energy Primer: A Handbook of Energy Market Basics. <https://www.ferc.gov/media/energy-primer-handbook-energy-market-basics> Accessed February 14th, 2024.

Turner, S.W., Voisin, N., Fazio, J., Hua, D. and Jourabchi, M., 2019. Compound climate events transform electrical power shortfall risk in the Pacific Northwest. *Nature Communications*, 10(1), p.8.

Actions taken: We have added to discussion a detailed explanation for not considering possible seasonal electricity demand patterns as a driver of annual CF trend (LINES 513-532).

R1-15: L242-245: I think it would not be that hard to check this given that EIA generation data are issued monthly. Even investigating a few of the big dams and whether their seasonal patterns have changed would be pretty important for claims I think.

Authors' response: We agree that seasonality of water availability is a potentially significant driver of annual CF decline. We also agree that our manuscript would be enhanced with some analysis on whether seasonal inflow patterns have changed at dams experiencing significant CF decline. The mechanism here would be intensification of the seasonal pattern of water availability causing increased spill on an annual basis (water availability overwhelming turbine capacity), or more frequent high inflow events causing similar behavior. An important point to note is that significant trend in seasonal flow regime does not necessarily mean that such change has driven CF decline. The reservoir will play a role in mitigating any effects of flow regime change. Nonetheless, the new analysis is worthwhile as it provides evidence that significant changes to the flow regime and flood magnitudes are uncommon across plants with significant CF change that cannot be attributed to change in annual water availability.

In relation to the suggestion to employ EIA hydropower generation records at monthly timescale, please note that while EIA-923 data are presented at a monthly resolution, most of these records (some 90% of plants covered) are collected as annual totals only. The EIA disaggregates to monthly using their own imputation method, rendering much of the monthly data inappropriate for studies requiring observation. Please see our recent work published in *Scientific Data* demonstrating significant inaccuracies with EIA-923 monthly data in most regions: <https://doi.org/10.1038/s41597-022-01748-x>.

Actions taken: We have added new analysis at the plant level investigating possible trends in peak flow conditions and seasonality of inflow conditions. This analysis is based on plants for which (a) CF has declined with statistical significance ($p < 0.05$), and (b) trend in annual water availability explains 60% or less of the CF decline observed (LINES 339-357; 743-749; Supplementary Information, Figure S3).

R1-12: L86: what does “climate-driven” mean here? Is that explicitly the impact of climate change, or a regional thing?

Authors' response: This section refers to the natural versus gauge-assimilated flow analysis. If we find that a trend in water availability over the study period appears in both natural and gauge-assimilated (non-natural) flow models, we know that any trend in the data derive from the climatic conditions (Pr, T) that force the model (rather than factors such as changes in water consumption and regulation, which would present in the gauge-assimilated data only).

Actions taken: Since our wording failed to make clear which part of the analysis this finding derives from, we have rewritten the sentence as: “*Analysis of natural versus gauge-assimilated flow at all plants reveals that the water availability trends linked to CF decline are more often caused by climatic factors (change in precipitation) than to non-climatic factors that reduce available water (e.g., land use change).*” (LINES 141-146)

R1-13: L90-93: *I think this is a much more measured way of presenting the results than is in the abstract.*

Authors' response: We agree.

Actions taken: We have amended the final sentence of the abstract (LINES 26-32).

R1-14: L97: *I think it would be helpful to discuss the populations of dams with + vs - CF in terms of total capacity and/or generation, in addition to number — e.g., it's important to know whether all the little dams are seeing declining CF while the bigger ones are mostly increasing when judging the influence of different drivers. I see you get into this a bit later but stating right up front "482 dams (X% of capacity, Y% of generation) would be valuable I think..*

Authors' response: We agree that this additional detail is helpful for readers.

Actions taken: We have added a sentence stating that plants with declining CF account for 85% of nameplate and 84% of generation across all studied plants (LINES 158-159).

R1-15: L126: *might be useful to note that the PNW dominates overall US hydro production, so we'd expect the trend to be similar for the country as a whole.*

Authors' response: Yes, PNW accounts for about a third of U.S. total generation, so certainly has a significant bearing trend at national scale.

Actions taken: We have a sentence to cover this point (LINES 195; 198-199).

R1-16: Fig2: *what's the time interval here (e.g., annual)? (suggest adding this to all relevant figures)*

Authors' response: Yes, all CF in this study is annual CF.

Actions taken: We have updated all figure captions to read "annual CF" in place of "CF".

R1-17: L186: *just flagging the paper inconsistently uses gauge vs gage (I'd probably go with gage)*

Authors' response: Well spotted, thank you!

Actions taken: Switched to “gauge” throughout.

R1-18: *L263: again here I think you can check this in your data. What are the generation trends?*

Authors’ response: This comment refers to our statement that capacity uprating would “increase average generation but may reduce capacity factor if water becomes constraining at higher ratings.” The text that follows this statement (“...we find scant evidence for capacity additions being a driver of CF trend...”) was in fact already based on analysis of the data, although results were not presented statistically. While poorly elaborated in our first submission, this analysis indicates a very limited role of capacity additions driving observed CF trend unexplained by change in water availability.

Actions taken: We have added new detail on our analysis of the possible impact of capacity additions, including assessment of the significance of shifts in mean CF corresponding to years of capacity upgrade (LINES 379-388; 740-742). Such shifts present with evidence greater than long-term linear trend in only six plants (see Supplementary Figure S5).

R1-19: *relatedly, EIA has age data and it wouldn’t be too hard to check whether there’s a big difference in CF trends for older dams (though sometimes major overhauls are tricky to spot).*

Authors’ response: We agree this should be checked. Based on an analysis using average generator age each plant (weighted by generator capacity), we find no pattern that would allow us attribute CF decline to plant age.

Actions taken: We have added new analysis showing CF trends as function of (1) mean (capacity-weighted) generator age at each plant; (2) FERC-license status; and (3) operating mode (peaking vs run-of-river) (LINES 411-414; Supplementary Information, Figure S4).

R1-19: *L288: I think it’s tricky to make the claim that this is “perhaps the most likely cause” based on your L298 and the fact that there are some checkable things about seasonality, plant age, generation trends, etc that would tell you how significant those factors are. The hydro literature tends to implicitly demonize environmental protections as a driver of reduced hydropower outputs and I think it’s important to be really careful not to fall into that without evidence, even if unintentionally. As written, the paper seems to pretty heavily imply that it’s environmental rules and other renewables that are driving these issues, but I don’t see a ton of evidence for that implication.*

Authors’ response: We are not aware of literature that implicitly or unfairly demonizes environmental protections as a driver of reduced hydropower output. We can confirm this is certainly not our intention. We agree that the evidence for one driver or another is not concrete enough to make strong assertions on the most likely cause of CF decline. The aim of our

discussion section is to weigh the available evidence for different drivers of CF decline and to offer some careful speculation on what we believe to be most important. Our analysis of one-off shift versus trend provides some evidence of abrupt operational change, and such change is not necessarily in support of environmental protections.

In response to the above comment and to Reviewer #3's suggestions relating to FERC-licensing and the Electric Consumers Protection Act of 1986 (see reviewer comments **R3-4** and **R3-5**) we have expanded our analysis and discussion with some evidence for FERC relicensing and associated operational changes being important. In response to comments **R1-15** and **R1-18** above we have also added analyses on hydrologic seasonality and plant age. While these analyses cannot conclusively rule out these factors as significant drivers of CF decline, it is our view that the weight of evidence points toward nonpower operations as being a very significant driver. This conclusion would also fit with our experiences and discussions with operators. However, we agree on the importance of not concluding strongly on the matter and of highlighting future research needs in this area.

Actions taken: We have added further caveats throughout discussion and in conclusions to convey to the reader that further research is needed to disentangle the various plausible drivers of CF decline (LINES 415; 548-549). In the abstract, we have switched “environmental” to “nonpower” to capture the range of possible operational changes that may have affected plant output (LINE 28). We have added analyses suggesting a rather limited role of other drivers (hydrologic change driving spill, generator age) (LINES 339-357, 743-749, Figure S3, Figure S4).

R1-20: *L303: maybe, but I think only if sustained: I'd actually expect mechanical issues to cause a pretty abrupt change.*

Authors' response: The nature of operational change, such as to comply with a FERC license renewal, is that it is sustained. In contrast, a mechanical issue causing outage or abrupt loss of efficiency would usually be fixed.

Actions taken: We have added a note to highlight that operational change to comply with FERC license renewals at hydropower dams are sustained (LINES 433-441).

R1-21: *L330: possibly? But is that super relevant in places with a lot of hydropower? I'm not convinced this is a huge issue in the PNW, for example, though issues like increased summer load due to population growth + increased need for cooling, which could throw off seasonal matches between hydro availability and power demand, could be important. Again I think this is pretty verifiable (check to see if big increases in wind/solar in places with a lot of hydro correlate with downward CF trends) and probably worth doing.*

Authors' response: This comment refers to our statement that emergence of wind and solar could be a driver of CF decline, owing to a shift toward flexible operations that may force hydro turbines to be operated at non-optimal ranges and with potentially increased spill that may reduce the overall output.

Interestingly, and in conflict with the comment above, Reviewer 2 asserts that wind and solar must be the primary driver of CF decline (see comment **R2-6**). We have thus decided to test this explicitly. If emergence of wind and solar is a primary driver of hydropower CF decline, then one would expect to see CF trends lessen in magnitude if analyzed over the period 1980–2009 (i.e., before wind or solar had been implemented at scale). We find that trends in this shortened period are not, in general, of lesser magnitude—confirming your suspicion that this is not a major driver.

Actions taken: We have added a new figure comparing trends in the overall study period (1980 – 2022) with a shortened period (1980 – 2009) (LINES 505-509; Supplementary Information Figure S1).

R1-22: *L382: interesting closing given that I feel one of the main args of the paper is that the decline is reversible since it's not mostly based on resource constraints?*

Authors' response: This comment refers to our closing comment that “*grid planning and electricity portfolio studies may need to accommodate the other drivers identified here to avoid overestimating hydropower’s future contribution to overall generation.*” While the paper indeed shows that much of the observed trend is not caused by resource constraints, we do not explicitly state that the decline is reversible. It may be reversed if: (1) there’s significant investment in generators and equipment; (2) there’s no further river regulation that encourages non-powered spill; (3) the hydropower fleet does not move toward becoming more of a flexible resource. Regardless, whether hydropower annual generation declines can be reversed or not, the existence and scale of decline that we observe—and its *potential* to continue into the future—ought to prompt a revision to the hydropower availability numbers adopted in grid planning and reliability studies.

Actions taken: We have added further commentary on reversibility of hydropower generation and CF decline (LINES 560-563).

Reviewer 2 – Balázs M. Fekete

R2-1: *This is the first time in my long history of reviewing to present my comments without reading the full paper but in the present case, I feel strongly that I don't need to read it any further.*

Authors' response: Thank you for signing your review, as it gave us the chance to re-read your recent work in this area and try to better understand your perspective. We suggested you as a reviewer, referencing your hydrological expertise as well as your recent work evaluating regional hydropower trends (Sharma et al., 2019). We think there has been some misinterpretation, and we've attempted to reconcile this in our responses below.

First, please note that our author team comprises hydropower engineers and researchers who interact with plant operators, utilities, and grid operators on a regular basis. We understand the unique nature of hydropower, its role in balancing variable resources (among other ancillary services), and its value to the grid—irrespective of its contribution to annual total generation. We also recognize that the nameplate capacity of a hydropower plant sometimes reflects design for high flow conditions, and that some hydropower plants operate as peaking plants.

We think that you must have misunderstood some critical components of our aim or approach to have produced such a dismissive review of this work. We regard our work to be both highly relevant and analytically sound, with two other expert reviewers in agreement. The misunderstanding of our work is evident in the assertions that we are “*lumping together capacity factors*” (we are not) and that our title implies “*another looming climate change-driven disaster*” (this is an inaccurate characterization of the title), as well as your comments that present our discussion of non-climatic drivers as somehow contrary to a premise or conclusion of the study. If you are asked to provide a further review, we ask that you please approach the study with a fresh mindset and without preconceptions on its aims or message.

For your convenience in finding our revisions in the revised manuscript, line numbers referenced below refer to the document *with* tracked changes.

R2-2: *I think the paper is fundamentally wrong in lumping together the capacity factors of different energy sources. The authors themselves acknowledges that peaking technology gas turbines has a meager 13% capacity factor, end yet they compare the hydropower capacity factors to other forms of energy such as solar or nuclear and discuss its decline in hydropower generation as if it was a negative trend that would limit the value of hydropower in the future.*

Authors' response: In case there is any ambiguity on the meaning of capacity factor, the definition employed in our study is consistent with US EIA: “The ratio of the electrical energy produced by a generating unit for the period of time considered to the electrical energy that could have been produced at continuous full power operation during the same period.” Capacity factor is not equal to the proportion of time the plant is running. It is used in our study as a metric of ***total annual generation*** that removes the effects of turbine upgrades and retirements. For plants

that have not changed in nameplate over the study period, the relative trend in annual capacity factor is *identical* to relative trend in annual generation. There is nothing unconventional or controversial about using capacity factor in this way. We think you may have misunderstood why we use annual capacity factor.

Our study is based on an analysis of annual capacity factor time series computed separately for hundreds of hydropower plants. The study does not lump together capacity factors of different energy sources. Capacity factors of other resources are not relevant to our analysis or findings. We do, in the opening paragraph only, *compare* different resource types according to their typical capacity factors. This is solely to highlight what distinguishes hydropower from other resources—to demonstrate to a reader unfamiliar with hydrology and reservoir operations that water availability and the rules surrounding water release and storage exert a major influence on hydropower generation, leading to capacity factors that are lower than one might expect given hydropower’s advantages of being readily dispatchable and with near-zero marginal production costs. On reading your recent paper, we see your emphasis on hydropower facilities having “intentionally low” capacity factor, since plant capacity is often designed for periods of high flow, or because hydropower plants often act partially as peaking plants (Fekete et al, 2023). We agree that this is one reason for the generally low CF of hydropower. However, our omission of this point in our introductory paragraph has zero bearing on our analysis or the findings of the study. Whether or not a hydropower plant acts as a peaking plant (about half do not), its annual output is a very important metric of its performance, and robust trend in its annual output signals important change worth documenting and exploring.

We are surprised that you feel comparing capacity factors of different resources warrants a dismissal of the study without reading beyond the introduction. In your earlier work assessing regional trend in US hydropower (Sharma et al., 2019), we note the following passage (p. 118):

“The capacity factor of conventional hydropower remains on the lower spectrum when compared among other resources. For example, within the first three months of 2018, capacity factors of hydropower were 44.9%, 48.8%, and 44.8% respectively. On the other hand, capacity factors of other resources were substantially higher, namely, municipal solid waste (72-76%), biomass including wood (52-55%) and geothermal (76-80%). Additionally, wind power is nearing the capacity factor of conventional hydropower (42-44%).”

To your second point (“... as if it was a negative trend that would limit the value of hydro in future”), there are many reasons why one could consider a decline in annual hydropower CF as a negative trend that limits the value of hydro relative to the past. Again, we are surprised that you think declining annual energy from hydropower is not something that may limit its future value, since your 2019 paper concludes that “...reservoir and dam-based hydroelectricity may not be an efficient energy resource... (and) perhaps it is timely to consider promoting other non-conventional renewable resources for energy production.” This because “...since 2000, the mean contribution of hydroelectricity has remained less than 10% of the total energy generated in the U.S. and has been declining since then.”

If declines in CF are caused by loss of storage due to sedimentation, impaired turbine efficiency, increased outage due to wear and tear, increased spill requirements, a climatic shift, etc., then the

plant operator is taking less revenue and hydropower is contributing less overall energy. This is obviously an undesirable outcome given hydropower is supplying cheap, low-carbon electricity. On the other hand, there are reasons why a decline in annual generation need not signal a loss of overall value in hydropower. One of these is operational shifts toward more flexibility to balance emerging wind and solar (trading overall output for enhanced flexibility). You mention this in your review as if it is not already an integral component of our study.

Sharma, S., Waldman, J., Afshari, S. and **Fekete, B.**, 2019. Status, trends and significance of American hydropower in the changing energy landscape. *Renewable and sustainable energy reviews*, 101, pp.112-122.

Fekete, B.M., Bacskó, M., Zhang, J. and Chen, M., 2023. Storage requirements to mitigate intermittent renewable energy sources: analysis for the US Northeast. *Frontiers in Environmental Science*.

Actions taken: We have rewritten the opening paragraph of the study, shifting the focus to potential drivers in trend annual energy generation and pointing out that such long-term trends have yet to be documented in the scientific literature (LINES 39-61).

R2-3: *The paper itself recognizes that the change in capacity factor of hydropower is not necessarily tied to the availability of water resources and noted that there are instances when the water resources increased and yet the capacity factor decreased due to the hydropower industry "sacrificing" portion of generation to support environmental objectives and more importantly "serving emerging grid needs".*

Authors' response: We do indeed recognize that long-term change in CF at a given plant is not necessarily tied to long-term change in water availability. In fact, we explicitly test the extent to which long-term change in CF is tied to long-term change in water availability. We do not understand why you raise this as if it undermines a premise or finding of our study. This comment suggests a misunderstanding of something very fundamental about our aim.

Actions taken: N/A

R2-4: *I think the authors should seriously revise their paper and reconsider what the "declining" capacity factor is meant to measure.*

Authors' response: We are measuring long-term change in annual generation across plants over the last 43 years. This is stated clearly in introduction. Our study is not dissimilar in motivation to your previous work that evaluates hydropower's overall contribution to annual energy at regional scales since 2000 (Sharma et al., 2019). We use annual capacity factor in place of generation because some plants have added or removed some capacity over the last 40 years. We want to analyze trends in hydropower generation without the conflating effects of plant upgrades or retirements. CF is simple and effective metric for this purpose. This is not an unconventional or controversial use of CF. Your report provides no clear explanation as to why analyzing CF trends at hydropower plants is problematic.

Actions taken: We have further emphasized the reasoning for using CF in our study (LINES 114-128).

R2-5: *Based on the title itself, one would think that decline is a sign of another "looming" climate change driven disaster, but when it comes to the capacity factor of hydro power vs. other forms of renewable energy sources there is a big difference.*

Authors' response: The title is: "Hydropower capacity factors trending down in the United States." It does not mention or implicate climate change. It does not mention looming disaster. It does not question the current or future value of the hydropower fleet. It makes no reference to other forms of energy. It simply states the key finding of the paper in a concise way, following the guidance set out by *Nature Communications*. Contrary to highlighting a climate change driven disaster, our paper demonstrates the limited role played by climate change in driving reductions of annual energy output from the nation's hydropower plants. Neither the word "disaster" nor any synonym for this word appears anywhere in our manuscript.

The study has nothing to do with capacity factors of other forms of renewable energy. We analyze hydropower only, and we measure, separately for more than 600 individual plants, how the capacity factor (and generation) has trended over 43 years.

Actions taken: N/A

R2-6: *Hydropower is often used for load balancing since this is the fastest means on the grid to add rapidly generation capacity. The "serving emerging grid needs" is likely the result of solar and wind flooding the grid necessitating hydropower intermittently kicking in as solar and wind abruptly drop out and turn off rapidly when solar or wind suddenly ramp up.*

Authors' response: We devote an entire section of our manuscript to non-climatic drivers of CF trend, including lengthy discussion on emergence of wind and solar and associated impact on the CF trends we observed (see LINES 476-512).

We believe you overstate the importance of wind and solar emergence in the long-term hydropower CF trends observed. The reason (as discussed in the paper and raised by Reviewer 1 – see comment **R1-21**) is that these technologies have emerged relatively recently, whereas our period of analysis goes back to 1980. One way of testing your hypothesis, however, is to see whether the trends are robust to removal of recent years during which wind and solar have emerged.

Actions taken: We have recalculated CF trends over the 30-year period 1980 – 2009 (i.e., before significant uptake of wind/solar) and added the comparison to Supplemental Information, with related commentary in our discussion section (LINES 5056-509; S.I. Figure S1). The analysis

confirms the very marginal role that solar/wind emergence may have played in observed long-term CF trends.

R2-7: *Just like the low capacity factor of peak generating gas turbine is the result of their ability to provide power when needed and stay idle when the demand decreases or the supply ramps up, the same is true for hydropower, so the decline have less or nothing to do with the availability of the resources, but utilizing its flexibility in providing power as needed. Low capacity factor of solar and wind is a measure of their failure to serve their purpose.*

Authors' response: Whether a hydropower plant is operated as a peaking plant or not, annual energy output from that plant, and long-term trends in annual energy output, remain an interesting and important metric for a variety of reasons. As reflected in Reviewer 3's insightful comment (**R3-5**), the hydropower community has long suspected that the introduction of the Electric Consumers Protection Act of 1986 resulted in impaired generation and CF at peaking plants. Our study offers new and compelling evidence of such declines across a large sample of plants. Furthermore, whether a plant is operated as a peaking plant or not, long-term change in water availability will certainly affect annual output. Peaker or not, almost no hydropower plants show annual output that is uncorrelated with annual reservoir inflow.

Incidentally, your description of hydropower plants as able to provide power when needed has been a lively subject of research in recent years. This is how hydropower has been represented in power system models that simulate generators dispatching to the grid (e.g., PLEXOS, GridView); one assigns the hydropower plant monthly energy budgets to reflect the amount of energy available, and the model dispatches power as needed to balance variable resources and promote an efficient and reliable power grid. Unfortunately, this representation of hydropower does not reflect reality. As discussed in detail by Marshall and Grubert (2022) as well as Rheinheimer et al. (2023)—and demonstrated through numerical experiments by Magee et al. (2022)—hydropower plants are rarely able to serve the electric grid in the way a gas peaking plant does. Most dams serve non-power objectives that dominate the release of water, such as complying with flood control rules, fish passage regulations, meeting downstream dissolved oxygen needs, and so on. Those constraints will require a certain volume of water released over a day, week, or month, and hydropower plant operators need operate within those boundaries. Many plants are unable participate whatsoever as flexible resources, due to downstream environmental, safety, amenity, and recreational risks associated with ramping up and down the release of water (Dworshak and Hungry Horse provide nice examples of this—see graph below and note Reviewer #3's comments on ramping restrictions in **R3-4**). Run-of-river facilities may have storage in the order of hours only and are therefore generating whatever portion of flow they can. Anything not turbined is spilled (meaning lost energy and lost revenue). See the example of Bonneville, below. Note that shaping the generation to meet load is often impossible in such systems, since there are multiple plants in cascade, and the power generated from one will dictate the when the flow arrives to the next (e.g., see the PNNL 2021 Hydropower Value Study showing how run of river facilities downstream of Grand Coulee are essentially generating whenever the outflows from the upstream projects arrive). Even plants that serve as peaking plants do not behave in the same fashion as a gas peaker. For example, see Libby below (a

peaking plant, according to the Existing Hydropower Assets database)—there’s somewhat of a diurnal ramp-up for daytime loads, but the plant is ramping from ~130MW to 220MW; the plant does not sit idle. Gas peaking plants don’t lose their fuel when not operating, whereas hydropower plants stand to lose their resource (via spill) if they don’t generate.

Figure 2 – Hourly generation data for four medium-to-large hydropower plants of different type, located in different parts of the BPA region (Bonneville – BON, Dworshak – DWR, Hungry Horse – HGH, Libby – LIB), covering four weeks of operation. Data generated using USACE Data Query; screenshot taken on 12-19-2023.

Marshall, A.M. and Grubert, E., 2022. Hydroelectricity Modeling for Low-Carbon and No-Carbon Grids: Empirical Operational Parameters for Optimization and Dispatch Models. *Earth's Future*, 10(8), p.e2021EF002503.

Rheinheimer, D.E., Tarroja, B., Rallings, A.M., Willis, A.D. and Viers, J.H., 2023. Hydropower representation in water and energy system models: a review of divergences and call for reconciliation. *Environmental Research: Infrastructure and Sustainability*.

Magee, T.M., Turner, S.W., Clement, M.A., Oikonomou, K., Zagona, E.A. and Voisin, N., 2022. Evaluating power grid model hydropower feasibility with a river operations model. *Environmental Research Letters*, 17(8), p.084035.

Actions taken: N/A

R2-8: Without recognizing these differences and evaluating whether the lower capacity factor arose from a necessity of load balancing or resulted from diminishing water resources the discussion of trends in capacity factors is either pointless or severely misleading.

I don't think this paper can be published without clarifying these differences, but I will be happy to read it carefully read it and provide constructive suggestions, once the authors addressed this fundamental comment.

Authors' response: Our study explicitly recognizes the importance of both resource constraints (water availability) and non-climatic drivers. We provide a novel analysis into the extent to which water availability trends have driven CF trends across hundreds of plants.

Actions taken: N/A

Reviewer 3

R3-Summary: *This work provides strong, well-supported evidence of a long-term decreasing trend in the capacity factor of the U.S. hydropower fleet and of that decrease being only partially attributable to changes in water availability. To my knowledge, this multi-decadal, plant-level analysis of U.S. hydropower capacity factor using statistical methods to perform attribution to water availability is novel. Improved understanding of capacity factor trends in the U.S. hydropower fleet and their drivers is of significance not only to the field of hydrology but to electricity grid models which sometimes make overly simplified assumptions about hydropower generation.*

The conclusions are properly supported with a combination of (a) thorough data analysis involving linkage of capacity, generation, and streamflow datasets and (b) simulation techniques and statistical tests. All the datasets used (EIA capacity and generation data, USGS' NHDPlus, ORNL's Dayflow) are publicly available with detailed methods to understand their scope and limitations. The use of natural and gauge-assimilated streamflow data to simulate capacity factor provides extra insight in that it allows separating changes in water availability due to climate versus other reasons such as reallocation of water to non-power purposes. The discussion of potential other drivers for the capacity factor declines is comprehensive and supported by some specific examples.

I commend the authors for the effort to produce high-quality visuals that combine maps and plots. These are a great way to summarize complex results where both spatial and time-series dimensions are of interest.

Authors' response: Thank you for the considerable time and effort put into conducting your review. We greatly appreciate and agree with all your suggestions. We have revised and enhanced the study accordingly.

For your convenience in finding our revisions in the revised manuscript, line numbers referenced below refer to the document *with* tracked changes.

R3-1: *Line 33: "hydropower plants operate with an annual CF of just 20-50% varying significantly by dam and year of operation". How comprehensive is the dataset used in the paper from which this statement is referenced? In my own analysis of capacity factor data combining EIA Form 923 and EIA Form 860, I have found instances of CFs higher than 50%. Often, they are small run-of-river plants which might not be part of the dataset in the referenced paper.*

Authors' response: Yes, you are correct. The 20-50% figure was an estimate from literature. Looking at the distribution across the 808 plants > 5MW, we see about 30% of plants have an average annual CF greater than 0.5, and across all plants and years we see that about 40% of data points have CF > 0.5.

Actions taken: The 20-50% estimate has now been removed with a rewrite of the opening paragraph (LINES 62-80).

R3-2: *Line 48: “Today in the United States, hydropower contributes approximately 7% of electricity generation and 38% of renewable generation.” The percentage of renewable electricity seems too high. I’ve seen numbers under 30% in recent years.*

Authors’ response: You’re right—and it depends on how many years we go back. Looking at the EIA data for 2013 – 2022 we conventional hydro contributing an average of 6.6% utility scale generation and 6.4% total electricity generation. It’s closer to 6% in the last three years. For the renewables, it’s about 40% if we look at the last ten years, but only 30% for the last three. We agree that the more recent data is most relevant given the rapid pace of change in renewables.

Actions taken: We’ve updated these numbers to 6% and 32%, respectively. We’ve also added that the data are based on 2020 – 2022 EIA information (LINES 82-83).

R3-3: *I suggest stating the percentage of U.S. capacity captured by the various subsets used in different stages of the analysis. Specifying the capacity percentage covered helps assess whether all findings can be interpreted as representative of the entire fleet. For instance, the 610 hydropower plants for which historical capacity factor data are available represent 90% of U.S. capacity. What percentage of capacity do the 362 plants for which gauge-assimilated streamflow data are available represent?*

Authors’ response: The 362 plants with a viable flow model contain many of the largest plants, so this sample accounts for 70% of total US installed conventional hydropower capacity.

Actions taken: We have added detail on representative capacity covered by the flow model (LINE 237-238). We’ve also made a correction on the % of US capacity represented by the sample of 610 plants (LINES 13, 118-119). This sample represents 90% of capacity of plants with nameplate > 5MW and 87% of capacity across all conventional hydro.

R3-4: *The initial discussion of observed capacity factor trends refers to differences by nameplate capacity (“declining capacity factor trends are particularly common at the largest plants by nameplate capacity”) and region (“declining hydropower CF appears throughout the nation but is most prevalent in the West”). I suggest exploring any noteworthy differences in trends along two other dimensions:*

1) Federal vs nonfederal (FERC-licensed) fleet

2) *Peaking versus run-of-river plants. I expect that decreased CF due to dam operational change relating to evolving environmental mitigation requirements might be more pronounced for peaking plants (e.g., ramping restrictions and restrictions in reservoir fluctuations added during relicensing).*

Authors' response: In terms of number of plants, we have a pretty even split between run-of-river, peaking, and other/unknown modes of operation across our sample (based on operational modes described in the Existing Hydropower Assets database). We find no strong evidence for stronger trend in any one operating category. However, your suggestion to compare FERC-licensed versus Federal yields an interesting result. FERC-licensed facilities account for about 70% of the 610 plants studied, but 92% (12 out of 13) of plants with CF trend < 15 PPPD. Median CF trend is also significantly lower across FERC-licensed facilities. This result lends some evidence to your next point on hydropower licensing, too.

Actions taken: We have added to Supplementary Information new analysis showing differences in trend along three dimensions: generator age (capacity-weighted mean at each plant), federal vs non-federal fleet, and mode of operation (Figure S4). This has been complemented with discussion in the main text (LINES 468-474).

R3-5: *In the discussion of operational changes as likely important drivers of declining CF, it might be worth to mention the Electric Consumers Protection Act of 1986. It required FERC to balance power and non-power values when issuing hydropower licenses. There is some evidence that projects relicensed post-ECPA, especially peaking projects, lost capacity during relicensing (due to increased weight given to environmental objectives). In their identification of one-time shifts in capacity factor, the authors state that the shifts do not neatly align with dam relicensing years. I suggest checking if the one-time shifts are more common for relicenses issued post-ECPA and if one-time shifts typically happen post-relicensing. It might take a few years for the licensees to implement the changes required in a relicense (especially if they involve capacity investments such as fish passage facilities) so operation shifts lagging relicensing a few years might still be driven by relicense conditions.*

Authors' response: Thank you for this insight on possible lagging between relicensing and implementation of new operations. Indeed, we find that permit issue years are quite close to the shifts identified in our CF data. Of the seven NY dams highlighted in Figure 5, six had a new permit issued within five years of the identified capacity shift. Of the five dams in the southeast region (AL-GE-SC) three dams show a shift within six years of the new permit.

Actions taken: We have added a passage on the import of the ECPA of 1986, citing a 1992 USGAO report highlighting some early indications of the impacts that new relicensing was having on generation (LINES 433-441). We have expanded our discussion on operational shifts, highlighting the possible causal relationship between relicense and CF shift at most of the non-federal dams mentioned in our shift analysis (LINES 457-465).

R3-6: ... *An additional potential driver of abrupt changes in operations are changes in the market setting/ structure in which the hydropower plant is operating. For instance, the Datta et al. (2022) paper referenced goes over changes in the operations of two hydropower plants after joining the Western Energy Imbalance Market. This is presented under the discussion of changes in operation due the needs of the power grid. However, evolving grid needs would likely show up gradually in the data while joining the WEIM (or another organized market with different bidding requirements/rules) can cause more abrupt changes.*

Authors' response: We agree.

Actions taken: We've added to our discussion on power grid needs to mention that this driver could present either as long-term change (slowly evolving operations as more wind/solar emerges) or abrupt change (entering a new market setting) (LINES 500-502).

R3-7: ... *In the conclusions or elsewhere in the discussion, I suggest mentioning that capacity factor is not the only metric that matters to evaluate the performance of a hydropower plant (regarding its power purpose). The availability of capacity at the times when it is most valuable and the flexibility to adjust output up or down with short notice are also important. Different drivers of capacity factor decline would affect dispatchability and flexibility differently which is another reason why further work to disentangle the relative importance of the various drivers proposed would be worthwhile.*

Authors' response: We agree on the importance of highlighting hydropower's continued importance to the grid irrespective of changes in annual output. We also agree on the need for future work to further understand how the drivers of CF decline have affected (and may continue to affect) dispatchability and flexibility.

Actions taken: We have rewritten the opening paragraph of introduction, highlighting hydropower's flexibility and ancillary services along with overall generation (LINES 40-61). We have also added to conclusions a passage on potential future work along the lines suggested (542-555).

R3-8: *I'm not a hydrologist and cannot comment on whether a different hydrology model would be better suited to analyze the data considered in this study. However, the authors provide a well-reasoned justification for their chosen methodology and it has been applied rigorously. The authors demonstrate improvements in fit for the chosen "new CF" model versus a simpler linear model of capacity factor.*

Authors' response: Thank you.

R3-9: ... *the authors provide a detailed description of the data used and equations estimated. Additionally, they state that they will make a repository available that will allow replication of the analysis upon publication.*

Authors' response: Thank you.

Actions taken: We have now added the link to our repository and data for reviewers who wish to examine our code and calculations (LINE 754).

Reviewers' Comments:

Reviewer #1:

Remarks to the Author:

Thanks to the authors for their thorough responses. I will still push back a little on the point about whether the CF declines are small — if the argument is that the headline number is a 13% cap-adjusted generation loss (over 30 years), that doesn't really refute the point that the rate of decline in CF is small — it's making a different point, that the accumulation of small CF declines has had a meaningful impact on generation. But I'd still argue that characterizing the main finding as a significant annual rate of CF decline is somewhat overstating the results. It looks like the authors made some revisions that do point the readers back to the generation trend rather than the CF trend (e.g. R1-4, R1-8), which I think addresses my point here.

And thanks — I'm very familiar with what capacity factor is. My point about "running less" applies both to "less frequently" and "less intensively." Your check about the nameplate changes is really what I was pointing toward here, so thanks for doing that. Based on some of the other comments (e.g., response to R1-10), I would still suggest that the authors go through and check really carefully that statements are not implicitly assuming no capacity changes (the author responses note in several places that CF and gen move together *when capacity doesn't change* but as you show, capacity has changed for a lot of dams over your analysis period) - e.g., the point that "Such a decrease in CF implies a ..." — as you note, that statement assumes no capacity changes, which isn't reflective of the system at hand. As you point out, the generation decline is 13%, not 23% as implied by the cap factor change, which I read as a pretty significant difference. As you point out, it's nontrivial, but it's not an huge drop. And again, I understand the difference between annual CF declines and cumulative gen declines, but characterizing an annual CF decline as large makes me assume the generation decline is *really* large. In general I would just encourage the authors to be really careful about how this is presented.

Overall, thank you for conducting the additional analyses, and I support publication.

Reviewer #2:

Remarks to the Author:

My comments in the first round of evaluation of this paper were admittedly harsh, but I am still convinced that it is important to set the stage properly when the objectives of a scientific paper are presented. I would like to note that in my previous comments, I did not reject the paper, but I wanted to express strongly that it is important to be clear that the capacity factor has different significance when it comes to hydropower generation.

I appreciate the clarifications that the author presented in their revised manuscript and this revision compelled me to fully read the paper this time and I can wholeheartedly support its publication. I don't think nitpicking of the paper in the second round of evaluation is helpful and fortunately, the other two reviewers apparently did a good job in paying attention to details.

This paper is undeniably an important work and highlight important changes in hydropower utilization. The presented analyses are sound and undoubtedly robust to support the conclusions of the paper. I support its publication in its current form.

Balázs Fekete
City College of New York

Reviewer #3:

Remarks to the Author:

The authors have sufficiently addressed all my initial comments and the revisions shown in the manuscript version with tracked changes overall strengthen the analysis.

I just have 4 minor comments on the revised version (line numbers correspond to the version with tracked changes) and recommend the manuscript for publication.

Line 46: Jacobsen et al., 2015.

This seems to be a typo. The reference list has this paper as Jacobson et al., 2015.

Line 125: Importantly, a decreasing trend in annual CF may not necessarily mean that a hydropower plant is running less.

I suggest adding "hours" at the end of the sentence for clarity.

Line 416: We find no evidence in our data to suggest a clear link between generator age and CF decline.

Based on my experience with hydropower generator age data, I know it is not always possible to distinguish between (a) age computed from the year the plant first came online and (b) age computed from the year the generator last underwent a major refurbishment/upgrade, which is a more meaningful piece of information as a proxy for asset health. It would be useful for the authors to give a bit more detail about the generator age data they use to help with interpretation of the above statement.

Line 516-535 (last paragraph before the Conclusions).

For hydropower generation (and CF) to be affected by demand patterns, it is not necessary for it to fully satisfy demand. If it is the marginal unit that is setting the price in a given market or market zone, the exact amount it generates in a given hour will depend on how much load there is. Despite its low marginal production costs, hydropower is sometimes the marginal unit (e.g., see Figure A-8 in NYISO 2022 State of the Market report: https://www.potomaceconomics.com/wp-content/uploads/2023/05/NYISO-2022-SOM-Full-Report__5-16-2023-final.pdf). Although I agree that changes in seasonal demand probably only explain a small portion of the capacity factor trend shown in the article, I think they should not be entirely dismissed as one of the contributing factors. Therefore, I suggest revising the wording in this paragraph to communicate that this can be a contributing factor to the generation trend but likely only explain a small portion of it.

Reviewer 1

R1-1: *Thanks to the authors for their thorough responses. I will still push back a little on the point about whether the CF declines are small — if the argument is that the headline number is a 13% cap-adjusted generation loss (over 30 years), that doesn't really refute the point that the rate of decline in CF is small — it's making a different point, that the accumulation of small CF declines has had a meaningful impact on generation. But I'd still argue that characterizing the main finding as a significant annual rate of CF decline is somewhat overstating the results. It looks like the authors made some revisions that do point the readers back to the generation trend rather than the CF trend (e.g. R1-4, R1-8), which I think addresses my point here.*

*And thanks — I'm very familiar with what capacity factor is. My point about “running less” applies both to “less frequently” and “less intensively.” Your check about the nameplate changes is really what I was pointing toward here, so thanks for doing that. Based on some of the other comments (e.g., response to R1-10), I would still suggest that the authors go through and check really carefully that statements are not implicitly assuming no capacity changes (the author responses note in several places that CF and gen move together *when capacity doesn't change* but as you show, capacity has changed for a lot of dams over your analysis period) - e.g., the point that “Such a decrease in CF implies a ...” — as you note, that statement assumes no capacity changes, which isn't reflective of the system at hand. As you point out, the generation decline is 13%, not 23% as implied by the cap factor change, which I read as a pretty significant difference. As you point out, it's nontrivial, but it's not an huge drop. And again, I understand the difference between annual CF declines and cumulative gen declines, but characterizing an annual CF decline as large makes me assume the generation decline is *really* large. In general I would just encourage the authors to be really careful about how this is presented.*

Overall, thank you for conducting the additional analyses, and I support publication.

Authors' response: Thank you again for your constructive comments and for supporting publication of our research.

Reviewer 2 – Balázs M. Fekete

R2-1: *My comments in the first round of evaluation of this paper were admittedly harsh, but I am still convinced that it is important to set the stage properly when the objectives of a scientific paper are presented. I would like to note that in my previous comments, I did not reject the paper, but I wanted to express strongly that it is important to be clear that the capacity factor has different significance when it comes to hydropower generation.*

I appreciate the clarifications that the author presented in their revised manuscript and this revision compelled me to fully read the paper this time and I can wholeheartedly support its publication. I don't think nitpicking of the paper in the second round of evaluation is helpful and fortunately, the other two reviewers apparently did a good job in paying attention to details.

This paper is undeniably an important work and highlight important changes in hydropower utilization. The presented analyses are sound and undoubtedly robust to support the conclusions of the paper. I support its publication in its current form.

Authors' response: Thank you for reconsidering our analysis and for the supportive comments.

Reviewer 3

R3-Summary: *The authors have sufficiently addressed all my initial comments and the revisions shown in the manuscript version with tracked changes overall strengthen the analysis. I just have 4 minor comments on the revised version (line numbers correspond to the version with tracked changes) and recommend the manuscript for publication.*

Authors' response: Thank you for these final iterations. We have amended the manuscript accordingly.

R3-1: *Line 46: Jacobsen et al., 2015. This seems to be a typo. The reference list has this paper as Jacobson et al., 2015.*

Actions taken: We have corrected this citation.

R3-2: *Line 125: Importantly, a decreasing trend in annual CF may not necessarily mean that a hydropower plant is running less. I suggest adding "hours" at the end of the sentence for clarity.*

Authors' response: We agree this was unclear.

Actions taken: We have changed to "...less often".

R3-3: *Line 416: We find no evidence in our data to suggest a clear link between generator age and CF decline. Based on my experience with hydropower generator age data, I know it is not always possible to distinguish between (a) age computed from the year the plant first came*

online and (b) age computed from the year the generator last underwent a major refurbishment/upgrade, which is a more meaningful piece of information as a proxy for asset health. It would be useful for the authors to give a bit more detail about the generator age data they use to help with interpretation of the above statement.

Authors' response: The available data are indeed based on when the generator became operational and do not include detail on any refurbishments that may have taken place.

Actions taken: We have amended our wording here to highlight that generator age does not inform on possible refurbishments.

R3-4: *Line 516-535 (last paragraph before the Conclusions). For hydropower generation (and CF) to be affected by demand patterns, it is not necessary for it to fully satisfy demand. If it is the marginal unit that is setting the price in a given market or market zone, the exact amount it generates in a given hour will depend on how much load there is. Despite its low marginal production costs, hydropower is sometimes the marginal unit (e.g., see Figure A-8 in NYISO 2022 State of the Market report: https://www.potomaceconomics.com/wp-content/uploads/2023/05/NYISO-2022-SOM-Full-Report_5-16-2023-final.pdf). Although I agree that changes in seasonal demand probably only explain a small portion of the capacity factor trend shown in the article, I think they should not be entirely dismissed as one of the contributing factors. Therefore, I suggest revising the wording in this paragraph to communicate that this can be a contributing factor to the generation trend but likely only explain a small portion of it.*

Authors' response: Thank you for highlighting this case.

Actions taken: We have amended the wording in this passage to inform on instances where hydropower is the marginal producer, adding that any seasonal demand shifts would be unlikely to explain *a significant portion* of long-term CF decline.

R3-5: *The code appears to make the full analysis pipeline reproducible and will be a usable resource for the community (but I did not download all the necessary data to reproduce the actual results and plots.*

Authors' response: Thank you. We will include in a data DOI both the code and the input data required to run the pipeline.

Actions taken: N/A